# Actor-Critic Approach for Temporal Predictive Clustering

## Abstract

Due to the wider availability of modern electronic health records (EHR), patient care data is often being stored in the form of time-series. Clustering such time-series data is crucial for patient phenotyping, anticipating patients' prognoses by identifying "similar" patients, and designing treatment guidelines that are tailored to homogeneous patient subgroups. In this paper, we develop a deep learning approach for clustering time-series data, where each cluster comprises patients who share similar future outcomes of interest (e.g., adverse events, the onset of comorbidities, etc.). The clustering is carried out by using our novel loss functions that encourage each cluster to have homogeneous future outcomes. We adopt actor-critic models to allow "back-propagation" through the sampling process that is required for assigning clusters to time-series inputs. Experiments on two real-world datasets show that our model achieves superior clustering performance over state-of-the-art benchmarks and identifies meaningful clusters that can be translated into actionable information for clinical decision-making.

## 1 Introduction

Chronic diseases – such as cystic fibrosis, dementia, and diabetes – are heterogeneous in nature, with widely differing outcomes even in narrow patient subgroups. Disease progression manifests through a broad spectrum of clinical factors, collected as a sequence of measurements over time in electronic health records (EHR), which gives a rise to complex progression patterns among patients (Samal et al., 2011). For example, cystic fibrosis evolves slowly, allowing for the development of related comorbidities and bacterial infections, and creating distinct behaviors/responses to therapeutic interventions, which in turn makes the survival and quality of life substantially different (Ramos et al., 2017). Identifying patient subgroups with similar progression patterns can be advantageous for understanding such heterogeneous underlying diseases. This allows clinicians to anticipate patients' prognoses by comparing "similar" patients in order to design treatment guidelines that are tailored to homogeneous patient subgroups (Zhang et al., 2019).

Temporal clustering has been recently used as a data-driven framework to partition patients with time-series observations into a set of clusters (i.e., into subgroups of patients). Recent research has typically focused on either finding fixed-length and low-dimensional representations (Zhang et al., 2019; Rusanov et al., 2016) or on modifying the similarity measure (Giannoula et al., 2018; Luong and Chandola, 2017) both in an attempt to apply the conventional clustering algorithms (e.g., $K$-means (Lloyd, 1982)) to time-series observations. However, clusters identified from these approaches these approaches are purely unsupervised – they do not account for each patient's observed outcome (e.g., adverse events, the onset of comorbidities, etc.) – which leads to heterogeneous clusters if the clinical presentation of the disease differs even for similar patients. Thus, a common prognosis in each cluster remains unknown which can mystify the understanding of the underlying disease progression (Boudier et al., 2019). For instance, patients who appear to have similar time-series observations may develop different sets of comorbidities in the future which, in turn, require different treatment guidelines to reduce such risks (Wami et al., 2013). To overcome this limitation, we focus on *predictive clustering* (Blockeel et al., 2017) which combines prediction with clustering. Therefore, the cluster assignments are optimized such that patients in a cluster share similar future outcomes to provide a prognostic value.

In this paper, we propose an actor-critic approach for temporal predictive clustering, which we call AC-TPC.[1] Our model consists of three neural networks – an *encoder*, a *selector*, and a *predictor* – and a set of centroid candidates. More specifically, the encoder maps an input time-series into a latent encoding; the selector utilizes the encoding and assigns a cluster to which the time-series belongs to via a sampling process; and the predictor estimates the future outcome distribution conditioned on either the encoding or the centroid of the selected cluster. The following three contributions render our model to identify the predictive clusters. First, to encourage each cluster to have homogeneous future outcomes, we define a clustering objective based on the Kullback-Leibler (KL) divergence between the predictor's output given the input time series, and the predictor's output given estimated cluster assignments. Second, we transform solving a combinatorial problem of identifying cluster into iteratively solving two sub-problems: optimization of the cluster assignments and optimization of the cluster centroids. Finally, we allow "back-propagation" through the sampling process of the selector by adopting the training of actor-critic models (Konda and Tsitsiklis, 2000).

Throughout the experiments, we show significant performance improvements over the state-of-the-art clustering methods on two real-world medical datasets. Then, to demonstrate the practical significance of our model, we consider a more realistic scenario where the future outcomes of interest are high-dimensional – such as, development of multiple comorbidities in the next year – and interpreting all possible combinations is intractable. Our experiments show that our model can identify meaningful clusters that can be translated into actionable information for clinical decision-making.

## 2 Problem Formulation

Let $\mathbf{X} \in \mathcal{X}$ and $Y \in \mathcal{Y}$ be random variables for an input feature and an output label (i.e., one or a combination of future outcome(s) of interest) with a joint distribution $p_{XY}$ (and marginal distributions are $p_X$ and $p_Y$, respectively) where $\mathcal{X}$ is the feature space and $\mathcal{Y}$ is the label space. Here, we focus our description on $C$-class classification tasks, i.e., $\mathcal{Y} = \{1, \cdots, C\}$.[2] We are given a time-series dataset $\mathcal{D} = \{(\mathbf{x}_t^n, y_t^n)_{t=1}^{T^n}\}_{n=1}^N$ comprising sequences of realizations (i.e., observations) of the pair $(\mathbf{X}, Y)$ for $N$ patients. Here, $(\mathbf{x}_t^n, y_t^n)_{t=1}^{T^n}$ is a sequence of $T^n$ observation pairs that correspond to patient $n$ and $t \in \mathcal{T}^n \triangleq \{1, \cdots, T^n\}$ denotes the time stamp at which the observations are made. From this point forward, we omit the dependency on $n$ when it is clear in the context and denote $\mathbf{x}_{1:t} = (\mathbf{x}_1, \cdots, \mathbf{x}_t)$ for ease of notation.

Our aim is to identify a set of $K$ *predictive clusters*, $\mathcal{C} = \{\mathcal{C}(1), \cdots, \mathcal{C}(K)\}$, for time-series data. Each cluster consists of homogeneous data samples, that can be represented by its centroid, based on a certain similarity measure. There are two main distinctions from the conventional notion of clustering. First, we treat subsequences of each times-series as data samples and focus on partitioning $\{\{\mathbf{x}_{1:t}^n\}_{t=1}^{T^n}\}_{n=1}^N$ into $\mathcal{C}$. Hence, we define a cluster as $\mathcal{C}(k) = \{\mathbf{x}_{1:t}^n | t \in \mathcal{T}^n, \ s_t^n = k\}$ for $k \in \mathcal{K} \triangleq \{1, \cdots, K\}$ where $s_t^n \in \mathcal{K}$ is the cluster assignment for a given $\mathbf{x}_{1:t}^n$. This is to flexibly update the cluster assignment (in real-time) to which a patient belongs as new observations are being accrued over time. Second, we define the similarity measure with respect to the label distribution and associate it with clusters to provide a prognostic value. More specifically, we want the distribution of output label for subsequences in each cluster to be homogeneous and, thus, can be well-represented by the centroid of that cluster. Let $S$ be a random variable for the cluster assignment – that depends on a given subsequence $\mathbf{x}_{1:t}$ – and $Y|S = k$ be a random variable for the output given cluster $k$. Then, such property of predictive clustering can be achieved by minimizing the following Kullback-Leibler (KL) divergence:

$$KL\big(Y_t|\mathbf{X}_{1:t} = \mathbf{x}_{1:t}\big\|Y_t|S_t = k\big) \quad \text{for } \mathbf{x}_{1:t} \in \mathcal{C}(k) \tag{1}$$

where $KL\big(Y_t|\mathbf{X}_{1:t} = \mathbf{x}_{1:t}\big\|Y_t|S_t = k\big) = \int_{y \in Y} p(y|\mathbf{x}_{1:t})\big(\log p(y|\mathbf{x}_{1:t}) - \log p(y|s_t)\big)dy$. Here, $p(y|\mathbf{x}_{1:t})$ and $p(y|s_t)$ are the label distributions conditioned on a subsequence $\mathbf{x}_{1:t}$ and a cluster assignment $s_t$, respectively. Note that (1) achieves its minimum when the two distributions are equivalent.

---

[1] Source code available at `https://github.com/ICLR2020-ACTPC/ACTPC_submission.git`

[2] In this paper, we focus our description on $C$-class classification task, i.e., $\mathcal{Y} = \{1, \cdots, C\}$; in Appendix A, we discuss simple modifications of our model for regression and $M$-dimensional binary classification tasks, i.e., $\mathcal{Y} = \mathbb{R}$ and $\mathcal{Y} = \{0, 1\}^M$, respectively.

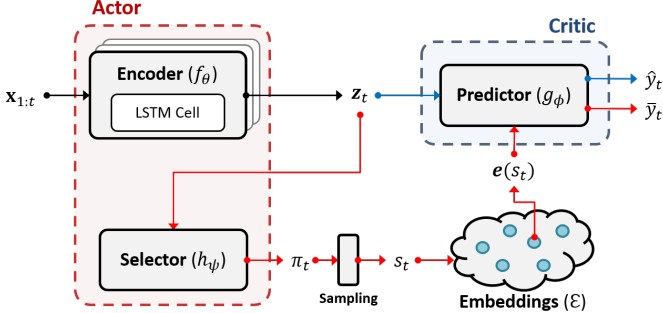

Figure 1: The block diagram of AC-TPC. The red line implies the procedure of estimating $p(y|S_t = s_t)$ which includes a sampling process and the blue line implies that of estimating $p(y|\mathbf{X}_{1:t} = \mathbf{x}_{1:t})$.

Finally, we establish our goal as identifying a set of predictive clusters $\mathcal{C}$ that optimizes the following objective:

$$\underset{\mathcal{C}}{\text{minimize}} \sum_{k \in \mathcal{K}} \sum_{\mathbf{x}_{1:t} \in \mathcal{C}(k)} KL\big(Y_t | \mathbf{X}_{1:t} = \mathbf{x}_{1:t} \big\| Y_t | S_t = k\big). \quad (2)$$

Unfortunately, the optimization problem in (2) is highly non-trivial. We need to estimate the objective function in (2) while solving a non-convex combinatorial problem of finding the optimal cluster assignments and cluster centroids.

## 3 ACTOR-CRITIC APPROACH FOR TEMPORAL PREDICTIVE CLUSTERING

To effectively estimate the objective function in (2), we introduce three networks – an *encoder*, a *selector*, and a *predictor* – and an *embedding dictionary* as illustrated in Figure 1. These components together provide the cluster assignment and the corresponding centroid based on a given sequence of observations and enable us to estimate the probability density $p(y|s_t)$. More specifically, we define each component as follows:

- The *encoder*, $f_\theta : \prod_{i=1}^{t} \mathcal{X} \to \mathcal{Z}$, is a RNN (parameterized by $\theta$) that maps a (sub)sequence of a time-series $\mathbf{x}_{1:t}$ to a latent representation (i.e., encoding) $\mathbf{z}_t \in \mathcal{Z}$ where $\mathcal{Z}$ is the latent space.

- The *selector*, $h_\psi : \mathcal{Z} \to \Delta^{K-1}$, is a fully-connected network (parameterized by $\psi$) that provides a probabilistic mapping to a categorical distribution from which the cluster assignment $s_t \in \mathcal{K}$ is being sampled.

- The *predictor*, $g_\phi : \mathcal{Z} \to \Delta^{C-1}$, is a fully-connected network (parameterized by $\phi$) that estimates the label distribution given the encoding of a time-series or the centroid of a cluster.

- The *embedding dictionary*, $\mathcal{E} = \{\mathbf{e}(1), \cdots, \mathbf{e}(K)\}$ where $\mathbf{e}(k) \in \mathcal{Z}$ for $k \in \mathcal{K}$, is a set of cluster centroids lying in the latent space which represents the corresponding cluster.

Here, $\Delta^{D-1} = \{\mathbf{q} \in [0,1]^K : q_1 + \cdots + q_D = 1\}$ is a $(D-1)$-simplex that denotes the probability distribution for a $D$-dimensional categorical (class) variable.

At each time stamp $t$, the *encoder* maps a input (sub)sequence $\mathbf{x}_{1:t}$ into a latent encoding $\mathbf{z}_t \triangleq f_\theta(\mathbf{x}_{1:t})$. Then, based on the encoding $\mathbf{z}_t$, the cluster assignment $s_t$ is drawn from a categorical distribution that is defined by the *selector* output, i.e., $s_t \sim Cat(\pi_t)$ where $\pi_t = [\pi_t(1), \cdots, \pi_t(K)] \triangleq h_\psi(\mathbf{z}_t)$. Once the assignment $s_t$ is chosen, we allocate the latent encoding $\mathbf{z}_t$ to an embedding $\mathbf{e}(s_t)$ in the *embedding dictionary* $\mathcal{E}$. Since the allocated embedding $\mathbf{e}(s_t)$ corresponds to the centroid of the cluster to which $\mathbf{x}_{1:t}$ belongs, we can, finally, estimate the density $p(y|s_t)$ in (2) as the output of the *predictor* given the embedding $\mathbf{e}(s_t)$, i.e., $\bar{y}_t \triangleq g_\phi(\mathbf{e}(s_t))$.

### 3.1 LOSS FUNCTIONS

In this subsection, we define loss functions to achieve our objective in (2); the details of how we train our model will be discussed in the following subsection.

**Predictive Clustering Loss:** Since finding the cluster assignment of a given sequence is a probabilistic problem due to the sampling process, the objective function in (2) must be defined as an expectation over the cluster assignment. Thus, we can estimate solving the objective problem in (2) as minimizing the following loss function:

$$\mathcal{L}_1(\theta, \psi, \phi, \mathcal{E}) = \mathbb{E}_{\mathbf{x}, y \sim p_{XY}} \left[ \sum_{t=1}^{T} \mathbb{E}_{s_t \sim Cat(\pi_t)} \left[ \ell_1(y_t, \bar{y}_t) \right] \right] \tag{3}$$

where $\ell_1(y_t, \bar{y}_t) = -\sum_{c=1}^{C} y_t^c \log \bar{y}_t^c$. Here, we slightly abuse the notation and denote $y = [y^1 \cdots y^C]$ as the one-hot encoding of $y$, and $y^c$ and $\bar{y}^c$ indicates the $c$-th component of $y$ and $\bar{y}$, respectively. It is worth to highlight that minimizing $\ell_1$ is equivalent to minimizing the KL divergence in (2) since the former term of the KL divergence is independent of our optimization procedure.

One critical question that may arise is how to avoid trivial solutions in this unsupervised setting of identifying the cluster assignments and the centroids (Yang et al., 2017). For example, all the embeddings in $\mathcal{E}$ may collapse into a single point or the selector simply assigns equal probability to all the clusters regardless of the input sequence. In both cases, our model will fail to correctly estimate $p(y|s_t)$ and, thus, end up finding a trivial solution. To address this issue, we introduce two auxiliary loss functions that are tailored to address this concern. It is worth to highlight that these loss functions are not subject to the sampling process and their gradients can be simply back-propagated.

**Sample-Wise Entropy of Cluster Assignment:** To motivate sparse cluster assignment such that the selector ultimately selects one dominant cluster for each sequence, we introduce sample-wise entropy of cluster assignment which is given as

$$\mathcal{L}_2(\theta, \psi) = \mathbb{E}_{\mathbf{x} \sim p_X} \left[ -\sum_{t=1}^{T} \sum_{k \in \mathcal{K}} \pi_t(k) \log \pi_t(k) \right] \tag{4}$$

where $\pi_t = [\pi_t(1) \cdots \pi_t(K)] = h_\psi(f_\theta(\mathbf{x}_{1:t}))$. The sample-wise entropy achieves its minimum when $\pi_t$ becomes an one-hot vector.

**Embedding Separation Loss:** To prevent the embeddings in $\mathcal{E}$ from collapsing into a single point, we define a loss function that encourages the embeddings to represent different label distributions, i.e., $g_\phi(\mathbf{e}(k))$ for $k \in \mathcal{K}$, from each other:

$$\mathcal{L}_3(\mathcal{E}) = -\sum_{k \neq k'} \ell_1(g_\phi(\mathbf{e}(k)), g_\phi(\mathbf{e}(k'))) \tag{5}$$

where $\ell_1$ is reused to quantify the distance between label distributions conditioned on each cluster. We minimize (5) when updating the embedding vectors $\mathbf{e}(1), \cdots, \mathbf{e}(K)$.

## 3.2 OPTIMIZATION

The optimization problem in (2) is a non-convex combinatorial problem because it comprises not only minimizing the KL divergence but also finding the optimal cluster assignments and centroids. Hence, we propose an optimization procedure that iteratively solves two subproblems: i) optimizing the three networks – the encoder, selector, and predictor – while fixing the embedding dictionary and ii) optimizing the embedding dictionary while fixing the three networks. Pseudo-code of AC-TPC can be found in Appendix F.

### 3.2.1 OPTIMIZING THE THREE NETWORKS – $f_\theta$, $h_\psi$, AND $g_\phi$

Finding predictive clusters incorporates the sampling process which is non-differentiable. Thus, to render "back-propagation", we utilize the training of actor-critic models (Konda and Tsitsiklis, 2000). More specifically, we view the combination of the encoder ($f_\theta$) and the selector ($h_\psi$) as the "actor" parameterized by $\omega_A = [\theta, \psi]$, and the predictor ($g_\phi$) as the "critic". The critic takes as input the the output of the actor (i.e., the cluster assignment) and estimates its value based on the sample-wise predictive clustering loss (i.e., $\ell_1(y_t, \bar{y}_t)$) given the chosen cluster. This, in turn, renders the actor to change the distribution of selecting a cluster to minimize such loss. Thus, it is important for the critic to perform well on the updated output of the actor while it is important for the actor

to perform well on the updated loss estimation. As such, the parameters for the actor and the critic need to be updated iteratively.

Given the embedding dictionary $\mathcal{E}$ fixed (thus, we will omit the dependency on $\mathcal{E}$), we train the actor, i.e., the encoder and the selector, by minimizing a combination of the predictive clustering loss $\mathcal{L}_1$ and the entropy of cluster assignments $\mathcal{L}_2$, which is given by

$$\mathcal{L}_A(\theta, \psi, \phi) = \mathcal{L}_1(\theta, \psi, \phi) + \alpha \mathcal{L}_2(\theta, \psi) \tag{6}$$

where $\alpha \geq 0$ is a coefficient chosen to balance between the two losses. To derive the gradient of this loss with respect $\omega_A = [\theta, \psi]$, we utilize the ideas from actor-critic models (Konda and Tsitsiklis, 2000) as follows; please refer to Appendix B for the detailed derivation:

$$\nabla_{\omega_A} \mathcal{L}_A(\theta, \psi, \phi) = \mathbb{E}_{\mathbf{x}, y \sim p_{XY}} \left[ \nabla_{\omega_A} \left( \sum_{t=1}^{T} \mathbb{E}_{s_t \sim Cat(\pi_t)} \big[ \ell_1(y_t, \bar{y}_t) \big] \right) \right] + \alpha \nabla_{\omega_A} \mathcal{L}_2(\theta, \psi)$$

$$= \mathbb{E}_{\mathbf{x}, y \sim p_{XY}} \left[ \sum_{t=1}^{T} \mathbb{E}_{s_t \sim Cat(\pi_t)} \big[ \ell_1(y_t, \bar{y}_t) \nabla_{\omega_A} \log \pi_t(s_t) \big] \right] + \alpha \nabla_{\omega_A} \mathcal{L}_2(\theta, \psi). \tag{7}$$

Note that since no sampling process is considered in $\mathcal{L}_2(\theta, \psi)$, we can simply derive $\nabla_{\omega_A} \mathcal{L}_2(\theta, \psi)$.

Iteratively with training the actor, we train the critic, i.e., the predictor, by minimizing the predictive clustering loss $\mathcal{L}_1$ as the following:

$$\mathcal{L}_C(\phi) = \mathcal{L}_1(\theta, \psi, \phi) \tag{8}$$

whose gradient with respect to $\phi$ can be givens as $\nabla_\phi \mathcal{L}_C(\phi) = \nabla_\phi \mathcal{L}_1(\theta, \psi, \phi)$. Note that since the critic is independent of the sampling process, the gradient can be simply back-propagated.

### 3.2.2 OPTIMIZING THE CLUSTER CENTROIDS

Now, once the parameters for the three networks $(\theta, \psi, \phi)$ are fixed (thus, we omit the dependency on $\theta$, $\psi$, and $\phi$), we updated the embeddings in $\mathcal{E}$ by minimizing a combination of the predictive clustering loss $\mathcal{L}_1$ and the embedding separation loss $\mathcal{L}_3$, which is given by

$$\mathcal{L}_E(\mathcal{E}) = \mathcal{L}_1(\mathcal{E}) + \beta \mathcal{L}_3(\mathcal{E}) \tag{9}$$

where $\beta \geq 0$ is a coefficient chosen to balance between the two losses.

### 3.2.3 INITIALIZING AC-TPC VIA PRE-TRAINING

Since we transform the non-trivial combinatorial optimization problem in (2) into iteratively solving two sub-problems, initialization is crucial to achieve better optimization as a similar concern has been addressed in (Yang et al., 2017).

Therefore, we initialize our model based on the following procedure. First, we pre-train the encoder and the predictor by minimizing the following loss function based on the predicted label distribution given the latent encodings of input sequences, i.e., $\hat{y}_t \triangleq g_\phi(\mathbf{z}_t) = g_\phi(f_\theta(\mathbf{x}_{1:t}))$, as the following:

$$\mathcal{L}_I(\theta, \phi) = \mathbb{E}_{\mathbf{x}, y \sim p_{XY}} \left[ - \sum_{t=1}^{T} \ell_1(y_t, \hat{y}_t) \right]. \tag{10}$$

Minimizing (10) encourages the latent encoding to be enriched with information for accurately predicting the label distribution. Then, we perform $K$-means (other clustering method can be also applied) based on the learned representations to initialize the embeddings $\mathcal{E}$ and the cluster assignments $\{\{s_t^n\}_{t=1}^{T^n}\}_{n=1}^{N}$. Finally, we pre-train the selector $h_\psi$ by minimizing the cross entropy treating the initialized cluster assignments as the true clusters.

## 4 RELATED WORK

Temporal clustering, also known as time-series clustering, is a process of unsupervised partitioning of the time-series data into clusters in such a way that homogeneous time-series are grouped together

based on a certain similarity measure. Temporal clustering is challenging because i) the data is often high-dimensional – it consists of sequences not only with high-dimensional features but also with many time points – and ii) defining a proper similarity measure for time-series is not straightforward since it is often highly sensitive to distortions (Ratanamahatana et al., 2005). To address these challenges, there have been various attempts to find a good representation with reduced dimensionality or to define a proper similarity measure for times-series (Aghabozorgi et al., 2015).

Recently, Baytas et al. (2017) and Madiraju et al. (2018) proposed temporal clustering methods that utilize low-dimensional representations learned by RNNs. These works are motivated by the success of applying deep neural networks to find "clustering friendly" latent representations for clustering static data (Xie et al., 2017; Yang et al., 2017). In particular, Baytas et al. (2017) utilized a modified LSTM auto-encoder to find the latent representations that are effective to summarize the input time-series and conducted $K$-means on top of the learned representations as an ad-hoc process. Similarly, Madiraju et al. (2018) proposed a bidirectional-LSTM auto-encoder that jointly optimizes the reconstruction loss for dimensionality reduction and the clustering objective. However, these methods do not associate a target property with clusters and, thus, provide little prognostic value in understanding the underlying disease progression.

Our work is most closely related to SOM-VAE (Fortuin et al., 2019). This method jointly optimizes a static variational auto-encoder (VAE), that finds latent representations of input features, and a self-organizing map (SOM), that allows to map the latent representations into a more interpretable discrete representations, i.e., the embeddings. However, there are three key differences between our work and SOM-VAE. First, SOM-VAE aims at minimizing the reconstruction loss that is specified as the mean squared error between the original input and the reconstructed input based on the corresponding embedding. Thus, similar to the aforementioned methods, SOM-VAE neither associates future outcomes of interest with clusters. In contrast, we focus on minimizing the KL divergence between the outcome distribution given the original input sequence and that given the corresponding embedding to build association between future outcomes of interest and clusters. Second, to overcome non-differentiability caused by the sampling process (that is, mapping the latent representation to the embeddings), Fortuin et al. (2019) applies the gradient copying technique proposed by (van den Oord et al., 2017), while we utilize the training of actor-critic model (Konda and Tsitsiklis, 2000). Finally, while we flexibly model time-series using LSTM, SOM-VAE handles time-series by integrating a Markov model in the latent representations. This can be a strict assumption especially in clinical settings where a patient's medical history is informative for predicting his/her future clinical outcomes (Ranganath et al., 2016).

## 5 EXPERIMENTS

In this section, we provide a set of experiments using two real-world time-series datasets. We iteratively update the three networks – the encoder, selector, and predictor – and the embedding dictionary as described in Section 3.2. For the network architecture, we constructed the encoder utilizing a single-layer LSTM (Hochreiter and Schmidhuber, 1997) with 50 nodes and constructed the selector and predictor utilizing two-layer fully-connected network with 50 nodes in each layer, respectively. The parameters $(\theta, \psi, \phi)$ are initialized by Xavier initialization (Glorot and Bengio, 2010) and optimized via Adam optimizer (Kingma and Ba, 2014) with learning rate of $0.001$ and keep probability $0.7$. We chose the balancing coefficients $\alpha, \beta \in \{0.1, 1.0, 3.0\}$ utilizing grid search that achieves the minimum validation loss in (3); the effect of different loss functions are further investigated in the experiments. Here, all the results are reported using 5 random 64/16/20 train/validation/test splits.

### 5.1 REAL-WORLD DATASETS

We conducted experiments to investigate the performance of AC-TPC on two real-world medical datasets; detailed statistics of each dataset can be found in Appendix C:

- **UK Cystic Fibrosis registry (UKCF)**[3]**:** This dataset records annual follow-ups for 5,171 adult patients (aged 18 years or older) enrolled in the UK CF registry over the period from 2008 and 2015, with a total of 25,012 hospital visits. Each patient is associated with 89 variables (i.e., 11 static and 78 time-varying features), including information on demographics and genetic

---

[3]https://www.cysticfibrosis.org.uk/the-work-we-do/uk-cf-registry

mutations, bacterial infections, lung function scores, therapeutic managements, and diagnosis on comorbidities. We set the development of different comorbidities in the next year as the label of interest at each time stamp.

- **Alzheimer's Disease Neuroimaging Initiative (ADNI)**[4]**:** This dataset consists of 1,346 patients in the Alzheimer's disease study with a total of 11,651 hospital visits, which tracks the disease progression via follow-up observations at 6 months interval. Each patient is associated with 21 variables (i.e., 5 static and 16 time-varying features), including information on demographics, biomarkers on brain functions, and cognitive test results. We set predictions on the three diagnostic groups – normal brain functioning, mild cognitive impairment, and Alzheimer's disease – as the label of interest at each time stamp.

## 5.2 BENCHMARKS

We compare AC-TPC with clustering methods ranging from conventional approaches based on $K$-means to the state-of-the-art approaches based on deep neural networks. All the benchmarks compared in the experiments are tailored to incorporate time-series data as described below:

- **Dynamic time warping followed by $K$-means**: Dynamic time warping (DTW) is utilized to quantify pairwise distance between two variable-length sequences and, then, $K$-means is applied (denoted as **KM-DTW**).

- **$K$-means with deep neural networks**: To handle variable-length time-series data, we utilize our encoder and predictor that are trained based on (10) for dimensionality reduction; this is to provide fixed-length and low-dimensional representations for time-series. Then, we apply $K$-means on the latent encodings $\mathbf{z}$ (denoted as **KM-E2P** ($\mathcal{Z}$)) and on the predicted label distributions $\hat{y}$ (denoted as **KM-E2P** ($\mathcal{Y}$)), respectively.

- **Extensions of DCN** (Yang et al., 2017): Since the DCN is designed for static data, we replace their static auto-encoder with a sequence-to-sequence network to incorporate time-series data (denoted as **DCN-S2S**).[5] In addition, to associated with the label distribution, we compare a DCN whose static auto-encoder is replaced with our encoder and predictor (denoted as **DCN-E2P**) to focus dimensionality reduction while preserving information for predicting the label.

- **SOM-VAE** (Fortuin et al., 2019): We compare with SOM-VAE – though, this method is oriented towards visualizing input data via SOM – since it naturally clusters time-series data (denoted as **SOM-VAE**). In addition, we compare with a variation of SOM-VAE by replacing the decoder with our predictor in order to find embeddings that capture information for predicting the label (denoted as **SOM-VAE-P**). For both cases, we set the dimension of SOM to $K$.

It is worth highlighting that the label information is provided for training DCN-E2P, KM-E2P, and SOM-VAE-P while the label information is not provided for training KM-DTW, DCN-S2S, and SOM-VAE. Please refer to Appendix D for the summary of major components of the tested benchmarks and the implementation details.

## 5.3 PERFORMANCE METRICS

To assess the clustering performance, we applied the following three standard metrics for evaluating clustering performances when the ground-truth cluster label is available: *purity score*, *normalized mutual information* (NMI) (Vinh et al., 2010), and *adjusted Rand index* (ARI) (Hubert and Arabie, 1985). More specifically, the purity score assesses how homogeneous each cluster is (ranges from 0 to 1 where 1 being a cluster consists of a single class), the NMI is an information theoretic measure of how much information is shared between the clusters and the labels that is adjusted for the number of clusters (ranges from 0 to 1 where 1 being a perfect clustering), and ARI is a corrected-for-chance version of the Rand index which is a measure of the percentage of correct cluster assignments (ranges from -1 to 1 where 1 being a perfect clustering and 0 being a random clustering). In a nutshell, all the three performance metrics are commonly used but all have its pros and cons; for instance, the

---

[4]https://adni.loni.usc.edu

[5]This extension is a representative of recently proposed deep learning approaches for clustering of both static data (Xie et al., 2017; Yang et al., 2017) and time-series data (Baytas et al., 2017; Madiraju et al., 2018) since these methods are built upon the same concept – that is, applying deep networks for dimensionality reduction to conduct conventional clustering methods, e.g., $K$-means.

Table 1: Performance Comparison on the UKCF and ADNI datasets.

| Dataset | Method | Purity | NMI | ARI | AUROC | AUPRC |
|---|---|---|---|---|---|---|
| UKCF | KM-DTW | $0.573\pm0.01^*$ | $0.010\pm0.01^*$ | $0.014\pm0.01^*$ | N/A | N/A |
| | KM-E2P ($\mathcal{Z}$) | $0.719\pm0.01^*$ | $0.211\pm0.01^*$ | $0.107\pm0.01^*$ | $0.726\pm0.01^*$ | $0.425\pm0.02^*$ |
| | KM-E2P ($\mathcal{Y}$) | $0.751\pm0.01^*$ | $0.325\pm0.01^*$ | $0.440\pm0.02^*$ | $0.807\pm0.00^*$ | $0.514\pm0.01^*$ |
| | DCN-S2S | $0.607\pm0.06^*$ | $0.059\pm0.08^*$ | $0.063\pm0.09^*$ | N/A | N/A |
| | DCN-E2P | $0.751\pm0.02^*$ | $0.275\pm0.02^*$ | $0.184\pm0.01^*$ | $0.772\pm0.03^*$ | $0.487\pm0.03^*$ |
| | SOM-VAE | $0.573\pm0.01^*$ | $0.006\pm0.00^*$ | $0.006\pm0.01^*$ | N/A | N/A |
| | SOM-VAE-P | $0.638\pm0.04^*$ | $0.201\pm0.05^*$ | $0.283\pm0.17^\dagger$ | $0.754\pm0.05^*$ | $0.331\pm0.07^*$ |
| | Proposed | $\mathbf{0.807\pm0.01}$ | $\mathbf{0.463\pm0.01}$ | $\mathbf{0.602\pm0.01}$ | $\mathbf{0.843\pm0.01}$ | $\mathbf{0.605\pm0.01}$ |
| ADNI | KM-DTW | $0.566\pm0.02^*$ | $0.019\pm0.02^*$ | $0.006\pm0.02^*$ | N/A | N/A |
| | KM-E2P ($\mathcal{Z}$) | $0.736\pm0.03^\dagger$ | $0.249\pm0.02$ | $0.230\pm0.03^\dagger$ | $0.707\pm0.01^*$ | $0.509\pm0.01$ |
| | KM-E2P ($\mathcal{Y}$) | $0.776\pm0.05$ | $0.264\pm0.07$ | $0.317\pm0.11$ | $0.756\pm0.04$ | $0.503\pm0.04$ |
| | DCN-S2S | $0.567\pm0.02^*$ | $0.005\pm0.00^*$ | $0.000\pm0.01^*$ | N/A | N/A |
| | DCN-E2P | $0.749\pm0.06$ | $0.261\pm0.05$ | $0.215\pm0.06^\dagger$ | $0.721\pm0.03^\dagger$ | $0.509\pm0.03$ |
| | SOM-VAE | $0.566\pm0.02^*$ | $0.040\pm0.06^*$ | $0.011\pm0.02^*$ | N/A | N/A |
| | SOM-VAE-P | $0.586\pm0.06^*$ | $0.085\pm0.08^*$ | $0.038\pm0.06^*$ | $0.597\pm0.10^\dagger$ | $0.376\pm0.05^*$ |
| | Proposed | $\mathbf{0.786\pm0.03}$ | $\mathbf{0.285\pm0.04}$ | $\mathbf{0.330\pm0.06}$ | $\mathbf{0.768\pm0.02}$ | $\mathbf{0.515\pm0.02}$ |

$*$ indicates $p$-value $< 0.01$,   $\dagger$ indicates $p$-value $< 0.05$

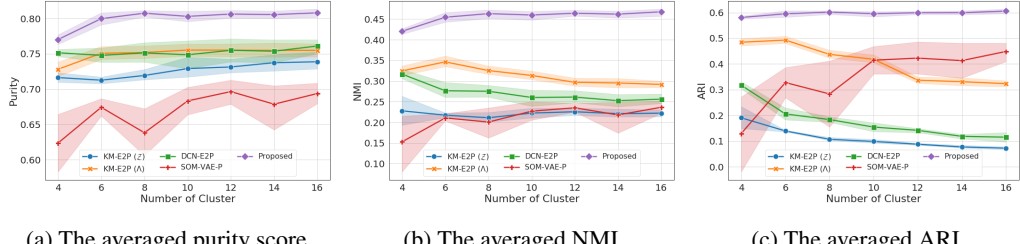

(a) The averaged purity score.     (b) The averaged NMI.     (c) The averaged ARI.

Figure 2: The purity score, NMI, and ARI (mean and 95% confidence interval) for the UKCF dataset ($C = 8$) with various $K$.

purity score easily converges to 1 when there are as many clusters as data samples. Thus, using them together suffices to demonstrate the effectiveness of the clustering methods.

To assess the prediction performance of the identified predictive clusters, we utilized both area under receiver operator characteristic curve (AUROC) and area under precision-recall curve (AUPRC) based on the label predictions of each cluster and the ground-truth binary labels on the future outcomes of interest. Note that the prediction performance is available only for the benchmarks that incorporate the label information during training.

### 5.4 Clustering Performance

We start with a simple scenario where the true class (i.e., the ground-truth cluster label) is available and the number of classes is tractable. In particular, we set $C = 2^3 = 8$ based on the binary labels for the development of three common comorbidities of cystic fibrosis – diabetes, ABPA, and intestinal obstruction – in the next year for the UKCF dataet and $C = 3$ based on the mutually exclusive three diagnostic groups for the ADNI dataset. We compare AC-TPC against the aforementioned benchmarks with respect to the clustering and prediction performance in Table 1.

As shown in Table 1, AC-TPC achieved performance gain over all the tested benchmarks in terms of both clustering and prediction performance – where most of the improvements were statistically significant with $p$-value $< 0.01$ or $p$-value $< 0.05$ – for both datasets. Importantly, clustering methods – i.e., KM-DTW, DCN-S2S, and SOM-VAE – that do not associate with the future outcomes of interest identified clusters that provide little prognostic value on the future outcomes (note that the true class is derived from the future outcome of interest). This is clearly shown by the ARI value near 0 which indicates that the identified clusters have no difference with random assignments. Therefore, similar sequences with respect to the latent representations tailored for reconstruction or with respect to the shape-based measurement using DTW can have very different future outcomes.

In Figure 2, we further investigate the purity score, NMI, and ARI by varying the number of clusters $K$ from 4 to 16 on the UKCF dataset in the same setting with that stated above (i.e., $C = 8$).

Here, the three methods – i.e., KM-DTW, DCN-S2S, and SOM-VAE – are excluded for better visualization. As we can see in Figure 2, our model rarely incur performance loss in both NMI and ARI while the benchmarks (except for SOM-VAE-P) showed significant decrease in the performance as $K$ increased (higher than $C$). This is because the number of clusters identified by AC-TPC (i.e., the number of activated clusters where we define cluster $k$ is activated if $|\mathcal{C}(k)| > 0$) was the same with $C$ most of the times, while the DCN-based methods identified exactly $K$ clusters (due to the $K$-means). Since the NMI and ARI are adjusted for the number of clusters, a smaller number of identified clusters yields, if everything being equal, a higher performance. In contrast, while our model achieved the same purity score for $K \geq 8$, the benchmark showed improved performance as $K$ increased since the purity score does not penalize having many clusters. This is an important property of AC-TPC that we do not need to know a priori what the number of cluster is which is a common practical challenge of applying the conventional clustering methods (e.g., $K$-means).

The performance gain of our model over SOM-VAE-P (and, our analysis is the same for SOM-VAE) comes from two possible sources: i) SOM-VAE-P mainly focuses on visualizing the input with SOM which makes both the encoder and embeddings less flexible – this is why it performed better with higher $K$ – and ii) the Markov property can be a too strict assumption for time-series data especially in clinical settings where a patient's medical history is informative for predicting his/her future clinical outcomes (Ranganath et al., 2016).

### 5.5 CONTRIBUTIONS OF THE AUXILIARY LOSS FUNCTIONS

As described in Section 3.1, we introduced two auxiliary loss functions – the sample-wise entropy of cluster assignment (4) and the embedding separation loss (5) – to avoid trivial solution that may arise in identifying the predictive clusters. To analyze the contribution of each auxiliary loss function, we report the average number of activated clusters, clustering performance, and prediction performance on the UKCF dataset with 3 comorbidities as described in Section 5.4. Throughout the experiment, we set $K = 16$ – which is larger than $C$ – to find the contribution of these loss functions to the number of activated clusters.

Table 2: Performance comparison with varying the balancing coefficients $\alpha, \beta$ for the UKCF dataset.

| Coefficients | | Clustering Performance | | | Prognostic Value | |
|---|---|---|---|---|---|---|
| $\alpha$ | $\beta$ | Activated No. | Purity | NMI | ARI | AUROC | AUPRC |
| 0.0 | 0.0 | 16 | 0.573±0.01 | 0.006±0.00 | 0.000±0.00 | 0.500±0.00 | 0.169±0.00 |
| 0.0 | 1.0 | 16 | 0.573±0.01 | 0.006±0.00 | 0.000±0.00 | 0.500±0.00 | 0.169±0.00 |
| 3.0 | 0.0 | 8.4 | 0.795±0.01 | 0.431±0.01 | 0.569±0.01 | 0.840±0.01 | 0.583±0.02 |
| 3.0 | 1.0 | 8 | **0.808±0.01** | **0.468±0.01** | **0.606±0.01** | **0.852±0.00** | **0.608±0.01** |

As we can see in Table 2, both auxiliary loss functions make important contributions in improving the quality of predictive clustering. More specifically, the sample-wise entropy encourages the selector to choose one dominant cluster. Thus, as we can see results with $\alpha = 0$, without the sample-wise entropy, our selector assigns an equal probability to all 16 clusters which results in a trivial solution. We observed that, by augmenting the embedding separation loss (5), AC-TPC identifies a smaller number of clusters owing to the well-separated embeddings; in Appendix E, we further investigate the usefulness of (5) in identifying the number of clusters in a data-driven fashion.

### 5.6 TARGETING MULTIPLE FUTURE OUTCOMES – A PRACTICAL SCENARIO

In this experiment, we focus on a more practical scenario where the future outcome of interest is high-dimensional and the number of classes based on all the possible combinations of future outcomes becomes intractable. For example, suppose that we are interested in the development of $M$ comorbidities in the next year whose possible combinations grow exponentially $C = 2^M$. Interpreting such a large number of patient subgroups will be a daunting task which hinders the understanding of underlying disease progression. Since different comorbidities may share common driving factors (Ronan et al., 2017), we hope our model to identify much smaller underlying (latent) clusters that govern the development of comorbidities. Here, to incorporate with $M$ comorbidities (i.e., $M$ binary labels), we redefine the output space as $\mathcal{Y} = \{0, 1\}^M$ and modify the predictor and loss functions, accordingly, as described in Appendix A.

Table 3: The top-3 frequent comorbidities developed in the next year for the 12 identified clusters. The values in parentheses indicate the corresponding frequency.

| Clusters | Top-3 Frequent Comorbidities | | |
|---|---|---|---|
| 0 | Diabetes (0.85) | Liver Enzymes (0.21) | Arthropathy (0.14) |
| 1 | Liver Enzymes (0.09) | Arthropathy (0.08) | Depression (0.07) |
| 2 | ABPA (0.77) | Osteopenia (0.21) | Intestinal Obstruction (0.11) |
| 3 | Asthma (0.89) | Liver Disease (0.87) | Diabetes (0.29) |
| 4 | Osteoporosis (0.76) | Diabetes (0.43) | Arthropathy (0.20) |
| 5 | Asthma (0.88) | Diabetes (0.81) | Osteopenia (0.28) |
| 6 | Liver Disease (0.85) | Asthma (0.03) | ABPA (0.09) |
| 7 | ABPA (0.83) | Diabetes (0.78) | Osteopenia (0.25) |
| 8 | Diabetes (0.94) | Liver Disease (0.83) | Liver Enzymes (0.43) |
| 9 | Asthma (0.89) | Osteopenia (0.26) | ABPA (0.19) |
| 10 | Osteopenia (0.82) | Diabetes (0.81) | Arthropathy (0.23) |
| 11 | Osteopenia (0.77) | Liver Enzymes (0.18) | Arthropathy (0.12) |

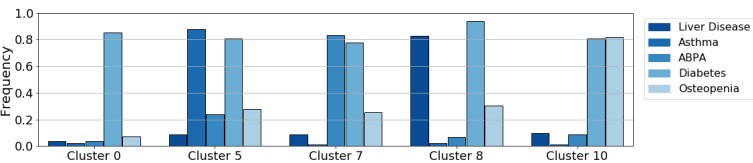

Figure 3: Clusters with high-risk of developing diabetes. We reported the cluster-specific frequencies of developing comorbidities – liver disease, asthma, ABPA, and osteopenia that are co-occurred with diabetes – in the next year.

Throughout this experiment, we aim at identifying subgroups of patients that are associated with the next-year development of 22 different comorbidities in the UKCF dataset. In Table 3, we reported 12 identified clusters – on average, the number of activated clusters were 13.6 – and the top three frequent comorbidities developed in the next year since the latest observation and the corresponding frequency; please refer to Appendix E for a full list. Here, the frequency is calculated in a cluster-specific fashion based on the true label. As we can see in Table 3, the identified clusters displayed very different label distributions; that is, the combination of comorbidities as well as their frequency were very different across the clusters. For example, patients in Cluster 1 experienced low-risk of developing any comorbities in the next year while 85% of patients in Cluster 0 experienced diabetes in the next year.

In Figure 3, we further investigated subgroups of patients – Cluster 0, 5, 7, 8, and 10 – who had high risk of developing diabetes in the next year. Although all these clusters displayed high risk of diabetes, the frequency of other co-occurred comorbidities was significantly different across the clusters. In particular, around 89% of the patients in Cluster 5 experienced asthma in the next year while it was less than 3% of the patients in the other cluster. Interestingly, "leukotriene" – a medicine commonly used to manage asthma – and "FEV$_1$% predicted" – a measure of lung function – were the two most different input features between patients in Cluster 5 and those in the other clusters. We observed similar findings in Cluster 7 with ABPA, Cluster 8 with liver disease, and Cluster 10 with osteopenia. Therefore, by grouping patients who are likely to develop a similar set of comorbidities, our method identified clusters that can be translated into actionable information for clinical decision-making.

## 6 CONCLUSION

In this paper, we introduced AC-TPC, a novel deep learning approach for predictive clustering of time-series data. We carefully defined novel loss functions to encourage each cluster to have homogeneous future outcomes (e.g., adverse events, the onset of comorbidities, etc.) and designed optimization procedures to address challenges and to avoid trivial solution in identifying such cluster assignments and the centroids. Throughout the experiments on two real-world datasets, we showed that our model achieves superior clustering performance over state-of-the-art and identifies meaningful clusters that can be translated into actionable information for clinical decision-making.

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

## A   VARIATIONS FOR REGRESSION AND BINARY CLASSIFICATION TASKS

As the task changes, estimating the label distribution and calculating the KL divergence in (2) must be redefined accordingly:

- For regression task, i.e., $\mathcal{Y} = \mathbb{R}$, we modify the predictor as $g_\phi : \mathcal{Z} \to \mathbb{R}$ and replace $\ell_1$ by $\ell_1(y_t, \bar{y}_t) = \|y_t - \bar{y}_t\|_2^2$. Minimizing $\ell_1(y_t, \bar{y}_t)$ is equivalent to minimizing the KL divergence between $p(y_t|\mathbf{x}_{1:t})$ and $p(y_t|s_t)$ when we assume these probability densities follow Gaussian distribution with the same variance.

- For the $M$-dimensional binary classification task, i.e., $\mathcal{Y} = \{0, 1\}^M$, we modify the predictor as $g_\phi : \mathcal{Z} \to [0, 1]^M$ and replace $\ell_1$ by $\ell_1(y_t, \bar{y}_t) = -\sum_{m=1}^M y_t^m \log \bar{y}_t^m + (1 - y_t^m) \log(1 - \bar{y}_t^m)$ which is required to minimize the KL divergence. Here, $y_t^m$ and $\bar{y}_t^m$ indicate the $m$-th element of $y_t$ and $\bar{y}_t$, respectively. The basic assumption here is that the distribution of each binary label is independent given the input sequence.

## B   DETAILED DERIVATION OF (7)

To derive the gradient of the predictive clustering loss in (7) with respect $\omega_A = [\theta, \psi]$, we utilized the ideas from actor-critic models (Konda and Tsitsiklis, 2000). The detailed derivation of the former term in (7) is described below (for notational simplicity, we omit the expectation on $\mathbb{E}_{\mathbf{x}, y \sim p_{XY}}$):

$$
\begin{aligned}
\nabla_\omega \left( \sum_{t=1}^T \mathbb{E}_{s_t \sim Cat(\pi_t)} \left[ \ell_1(y_t, \bar{y}_t) \right] \right) &= \nabla_\omega \left( \sum_{t=1}^T \sum_{s_t \in \mathcal{K}} \pi_t(s_t) \ell_1(y_t, \bar{y}_t) \right) \\
&= \sum_{t=1}^T \sum_{s_t \in \mathcal{K}} \nabla_\omega \pi_t(s_t) \ell_1(y_t, \bar{y}_t) \\
&= \sum_{t=1}^T \sum_{s_t \in \mathcal{K}} \frac{\nabla_\omega \pi_t(s_t)}{\pi_t(s_t)} \pi_t(s_t) \ell_1(y_t, \bar{y}_t) \qquad (11) \\
&= \sum_{t=1}^T \sum_{s_t \in \mathcal{K}} \pi_t(s_t) \ell_1(y_t, \bar{y}_t) \nabla_\omega \log \pi_t(s_t) \\
&= \sum_{t=1}^T \mathbb{E}_{s_t \sim Cat(\pi_t)} \left[ \ell_1(y_t, \bar{y}_t) \nabla_\omega \log \pi_t(s_t) \right]
\end{aligned}
$$

## C   DETAILS ON THE DATASETS

### C.1   UKCF DATASET

UK Cystic Fibrosis registry (UKCF)[6] records annual follow-ups for 5,171 adult patients (aged 18 years or older) over the period from 2008 and 2015, with a total of 25,012 hospital visits. Each patient is associated with 89 variables (i.e., 11 static and 78 time-varying features), including information on demographics and genetic mutations, bacterial infections, lung function scores, therapeutic managements, and diagnosis on comorbidities. The detailed statistics are given in Table 4.

### C.2   ADNI DATASET

Alzheimer's Disease Neuroimaging Initiative (ADNI)[7] study consists of 1,346 patients with a total of 11,651 hospital visits, which tracks the disease progression via follow-up observations at 6 months interval. Each patient is associated with 21 variables (i.e., 5 static and 16 time-varying features), including information on demographics, biomarkers on brain functions, and cognitive test results.

---

[6] https://www.cysticfibrosis.org.uk/the-work-we-do/uk-cf-registry
[7] https://adni.loni.usc.edu

Table 4: Summary and description of the UKCF dataset.

**STATIC COVARIATES**

| | | Type | Freq. | | Type | Freq. |
|---|---|---|---|---|---|---|
| Demographic | Gender | Bin. | 0.55 | | | |
| Genetic | Class I Mutation | Bin. | 0.05 | Class VI Mutation | Bin. | 0.86 |
| | Class II Mutation | Bin. | 0.87 | DF508 Mutation | Bin. | 0.87 |
| | Class III Mutation | Bin. | 0.89 | G551D Mutation | Bin. | 0.06 |
| | Class IV Mutation | Bin. | 0.05 | Homozygous | Bin. | 0.58 |
| | Class V Mutation | Bin. | 0.04 | Heterozygous | Bin | 0.42 |

**TIME-VARYING COVARIATES**

| | | Type | Mean (Freq.) | Min / Max | | Type | Mean (Freq.) | Min / Max |
|---|---|---|---|---|---|---|---|---|
| Demographic | Age | Cont. | 30.4 | 18.0 / 86.0 | Height | Cont. | 168.0 | 129.0 / 198.6 |
| | Weight | Cont. | 64.1 | 24.0 / 173.3 | BMI | Cont. | 22.6 | 10.9 / 30.0 |
| | Smoking Status | Bin. | 0.1 | | | | | |
| Lung Func. Scores | $FEV_1$ | Cont. | 2.3 | 0.2 / 6.3 | Best $FEV_1$ | Cont. | 2.5 | 0.3 / 8.0 |
| | $FEV_1$ % Pred. | Cont. | 65.1 | 9.0 / 197.6 | Best $FEV_1$ % Pred. | Cont. | 71.2 | 7.5 / 164.3 |
| Hospitalization | IV ABX Days Hosp. | Cont. | 12.3 | 0 / 431 | Non-IV Hosp. Adm. | Cont. | 1.2 | 0 / 203 |
| | IV ABX Days Home | Cont. | 11.9 | 0 / 441 | | | | |
| Lung Infections | B. Cepacia | Bin. | 0.05 | | P. Aeruginosa | Bin. | 0.59 | |
| | H. Influenza | Bin. | 0.05 | | K. Pneumoniae | Bin. | 0.00 | |
| | E. Coli | Bin. | 0.01 | | ALCA | Bin. | 0.03 | |
| | Aspergillus | Bin. | 0.14 | | NTM | Bin. | 0.03 | |
| | Gram-Negative | Bin. | 0.01 | | Xanthomonas | Bin. | 0.05 | |
| | S. Aureus | Bin. | 0.30 | | | | | |
| Comorbidities | Liver Disease | Bin. | 0.16 | | Depression | Bin. | 0.07 | |
| | Asthma | Bin. | 0.15 | | Hemoptysis | Bin. | 0.01 | |
| | ABPA | Bin. | 0.12 | | Pancreatitus | Bin. | 0.01 | |
| | Hypertension | Bin. | 0.04 | | Hearing Loss | Bin. | 0.03 | |
| | Diabetes | Bin. | 0.28 | | Gall bladder | Bin. | 0.01 | |
| | Arthropathy | Bin. | 0.09 | | Colonic structure | Bin. | 0.00 | |
| | Bone fracture | Bin. | 0.01 | | Intest. Obstruction | Bin. | 0.08 | |
| | Osteoporosis | Bin. | 0.09 | | GI bleed – no var. | Bin. | 0.00 | |
| | Osteopenia | Bin. | 0.21 | | GI bleed – var. | Bin. | 0.00 | |
| | Cancer | Bin. | 0.00 | | Liver Enzymes | Bin. | 0.16 | |
| | Cirrhosis | Bin. | 0.03 | | Kidney Stones | Bin. | 0.02 | |
| Treatments | Dornase Alpha | Bin. | 0.56 | | Inhaled B. BAAC | Bin. | 0.03 | |
| | Anti-fungals | Bin. | 0.07 | | Inhaled B. LAAC | Bin. | 0.08 | |
| | HyperSaline | Bin. | 0.23 | | Inhaled B. SAAC | Bin. | 0.05 | |
| | HypertonicSaline | Bin. | 0.01 | | Inhaled B. LABA | Bin. | 0.11 | |
| | Tobi Solution | Bin. | 0.20 | | Inhaled B. Dilators | Bin. | 0.57 | |
| | Cortico Combo | Bin. | 0.41 | | Cortico Inhaled | Bin. | 0.15 | |
| | Non-IV Ventilation | Bin. | 0.05 | | Oral B. Theoph. | Bin. | 0.04 | |
| | Acetylcysteine | Bin. | 0.02 | | Oral B. BA | Bin. | 0.03 | |
| | Aminoglycoside | Bin. | 0.03 | | Oral Hypo. Agents | Bin. | 0.01 | |
| | iBuprofen | Bin. | 0.00 | | Chronic Oral ABX | Bin. | 0.526 | |
| | Drug Dornase | Bin. | 0.02 | | Cortico Oral | Bin. | 0.14 | |
| | HDI Buprofen | Bin. | 0.00 | | Oxygen Therapy | Bin. | 0.11 | |
| | Tobramycin | Bin. | 0.03 | | $O_2$ Exacerbation | Bin. | 0.03 | |
| | Leukotriene | Bin. | 0.07 | | $O_2$ Nocturnal | Bin. | 0.03 | |
| | Colistin | Bin. | 0.03 | | $O_2$ Continuous | Bin. | 0.03 | |
| | Diabetes Insulin | Bin. | 0.01 | | $O_2$ Pro re nata | Bin. | 0.01 | |
| | Macrolida ABX | Bin. | 0.02 | | | | | |

ABX: antibiotics

Table 5: Summary and description of the ADNI dataset.

**STATIC COVARIATES**

| | | Type | Mean (Freq.) | Min/Max (Mode) | | Type | Mean (Freq.) | Min/Max (Mode) |
|---|---|---|---|---|---|---|---|---|
| Demographic | Race | Cat. | 0.93 | White | Ethnicity | Cat. | 0.97 | No Hisp/Latino |
| | Education | Cat. | 0.23 | C16 | Marital Status | Cat. | 0.75 | Married |
| Genetic | $APOE_4$ | Cont. | 0.44 | 0/2 | | | | |

**TIME-VARYING COVARIATES**

| | | Type | Mean | Min / Max | | Type | Mean | Min / Max |
|---|---|---|---|---|---|---|---|---|
| Demographic | Age | Cont. | 73.6 | 55/92 | | | | |
| Biomarker | Entorhinal | Cont. | 3.6E+3 | 1.0E+3 / 6.7E+3 | Mid Temp | Cont. | 2.0E+4 | 8.9E+3 / 3.2E+4 |
| | Fusiform | Cont. | 1.8E+5 | 9.0E+4 / 2.9E+5 | Ventricles | Cont. | 4.1E+4 | 5.7E+3 / 1.6E+5 |
| | Hippocampus | Cont. | 6.9E+3 | 2.8E+3 / 1.1E+4 | Whole Brain | Cont. | 1.0E+6 | 6.5E+5 / 1.5E+6 |
| | Intracranial | Cont. | 1.5E+6 | 2.9E+2 / 2.1E+6 | | | | |
| Cognitive | ADAS-11 | Cont. | 8.58 | 0/70 | ADAS-13 | Cont. | 13.61 | 0/85 |
| | CRD Sum of Boxes | Cont. | 1.21 | 0/17 | Mini Mental State | Cont. | 27.84 | 2/30 |
| | RAVLT Forgetting | Cont. | 4.19 | -12/15 | RAVLT Immediate | Cont. | 38.25 | 0/75 |
| | RAVLT Learning | Cont. | 4.65 | -5/14 | RAVLT Percent | Cont. | 51.70 | -500/100 |

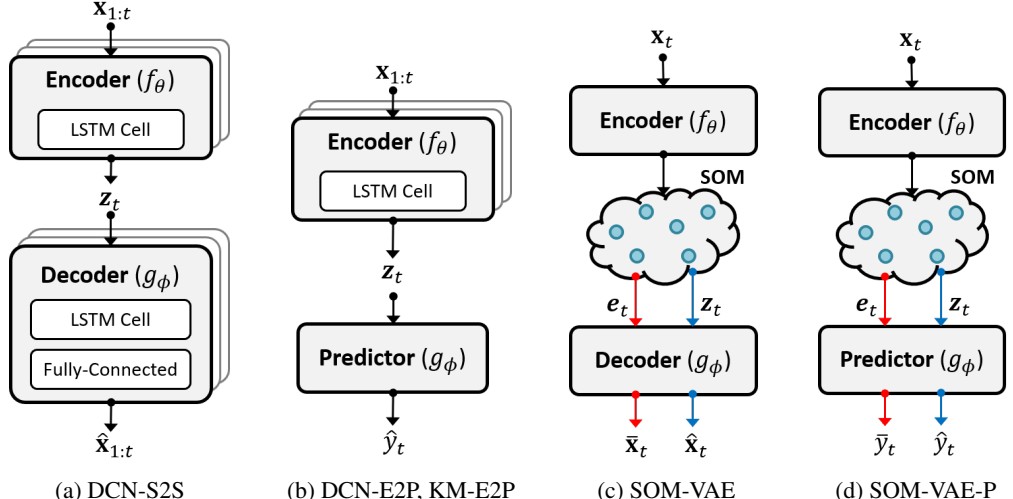

Figure 4: The block diagrams of the tested benchmarks.

Table 6: Comparison table of benchmarks.

| Methods | Handling Time-Series | Clustering Method | Similarity Measure | Label Provided | Label Associated |
|---|---|---|---|---|---|
| KM-DTW | DTW | $K$-means | DTW | N | N |
| KM-E2P ($\mathcal{Z}$) | RNN | $K$-means | Euclidean in $\mathcal{Z}$ | Y | Y (indirect) |
| KM-E2P ($\mathcal{Y}$) | RNN | $K$-means | Euclidean in $\mathcal{Y}$ | Y | Y (direct) |
| DCN-S2S | RNN | $K$-means | Euclidean in $\mathcal{Z}$ | N | N |
| DCN-E2P | RNN | $K$-means | Euclidean in $\mathcal{Z}$ | Y | Y (indirect) |
| SOM-VAE | Markov model | embedding mapping | reconstruction loss | N | N |
| SOM-VAE-P | Markov model | embedding mapping | prediction loss | Y | Y (direct) |
| Proposed | RNN | embedding mapping | KL divergence | Y | Y (direct) |

The three diagnostic groups were normal brain functioning (0.55), mild cognitive impairment (0.43), and Alzheimer's disease (0.02). The detailed statistics are given in Table 5.

## D  DETAILS ON THE BENCHMARKS

We compared AC-TPC in the experiments with clustering methods ranging from conventional approaches based on $K$-means to the state-of-the-art approaches based on deep neural networks. The details of how we implemented the benchmarks are described as the following:

- **Dynamic time warping followed by $K$-means**[8]: Dynamic time warping (DTW) is utilized to quantify pairwise distance between two variable-length sequences and, then, $K$-means is applied (denoted as **KM-DTW**).

- **$K$-means with deep neural networks**: To handle variable-length time-series data, we utilized an encoder-predictor network as depicted in Figure 4b and trained the network based on (10) for dimensionality reduction; this is to provide fixed-length and low-dimensional representations for time-series. Then, we applied $K$-means on the latent encodings **z** (denoted as **KM-E2P ($\mathcal{Z}$)**) and on the predicted label distributions $\hat{y}$ (denoted as **KM-E2P ($\mathcal{Y}$)**), respectively. We implemented the encoder and predictor of KM-E2P with the same network architectures with those of our model: the encoder is a single-layer LSTM with 50 nodes and the decoder is a two-layered fully-connected network with 50 nodes in each layer.

- **Extensions of DCN**[9] (Yang et al., 2017): Since the DCN is designed for static data, we utilized a sequence-to-sequence model in Figure 4a for the encoder-decoder network as an extension to

---

[8]https://github.com/rtavenar/tslearn
[9]https://github.com/boyangumn/DCN

incorporate time-series data (denoted as **DCN-S2S**) and trained the network based on the reconstruction loss (using the reconstructed input sequence $\hat{\mathbf{x}}_{1:t}$). For implementing DCN-S2S, we used a single-layer LSTM with 50 nodes for both the encoder and the decoder. And, we augmented a fully-connected layer with 50 nodes is used to reconstruct the original input sequence from the latent representation of the decoder.

In addition, since predictive clustering is associated with the label distribution, we compared a DCN whose encoder-decoder structure is replaced with our encoder-predictor network in Figure 4b (denoted as **DCN-E2P**) to focus the dimensionality reduction – and, thus, finding latent encodings where clustering is performed – on the information for predicting the label distribution. We implemented the encoder and predictor of DCN-E2P with the same network architectures with those of our model as described in Section 5.

- **SOM-VAE**[10] (Fortuin et al., 2019): We compare with SOM-VAE – though, this method is oriented towards visualization of input data via SOM – since it naturally clusters time-series data assuming Markov property (denoted as **SOM-VAE**). We replace the original CNN architecture of the encoder and the decoder with three-layered fully-connected network with 50 nodes in each layer, respectively. The network architecture is depicted in Figure 4c where $\hat{\mathbf{x}}_t$ and $\bar{\mathbf{x}}_t$ indicate the reconstructed inputs based on the encoding $\mathbf{z}_t$ and the embedding $\mathbf{e}_t$ at time $t$, respectively.

  In addition, we compare with a variation of SOM-VAE by replacing the decoder with the predictor to encourage the latent encoding to capture information for predicting the label distribution (denoted as **SOM-VAE-P**). For the implementation, we replaced the decoder of SOM-VAE with our predictor which is a two-layered fully-connected layer with 50 nodes in each layer to predict the label distribution as illustrated in Figure 4d. Here, $\hat{y}_t$ and $\bar{y}_t$ indicate the predicted labels based on the encoding $\mathbf{z}_t$ and the embedding $\mathbf{e}_t$ at time $t$, respectively.

  For both cases, we used the default values for balancing coefficients of SOM-VAE and the dimension of SOM to be equal to $K$.

We compared and summarized major components of the benchmarks in Table 6.

# E ADDITIONAL EXPERIMENTS

## E.1 ADDITIONAL RESULTS ON TARGETING MULTIPLE FUTURE OUTCOMES

Throughout the experiment in Section 5.6, we identified 12 subgroups of patients that are associated with the next-year development of 22 different comorbidities in the UKCF dataset. In Table 7, we reported 12 identified clusters and the full list of comorbidities developed in the next year since the latest observation and the corresponding frequency. Here, the frequency is calculated in a cluster-specific fashion based on the true label.

## E.2 TRADE-OFF BETWEEN CLUSTERING AND PREDICTION PERFORMANCE

In predictive clustering, the trade-off between the clustering performance (for better interpretability) – which quantifies how the data samples are homogeneous within each cluster and heterogeneous across clusters with respect to the future outcomes of interest – and the prediction performance is a common issue. The most important parameter that governs this trade-off is the number of clusters. More specifically, increasing the number of clusters will make the predictive clusters have higher diversity to represent the output distribution and, thus, will increase the prediction performance while decreasing the clustering performance. One extreme example is that there are as many clusters as data samples which will make the identified clusters fully individualized; as a consequence, each cluster will lose the interpretability as it no longer groups similar data samples.

To highlight this trade-off, we conducted experiments under the same experimental setup with that of Section 5.6 where our aim is to identify underlying (unknown) clusters when the future outcome of interest is high-dimensional. For the performance measures, we utilized the AUROC and AUPRC to assess the prediction performance, and utilized the average Silhouette index (SI) (Rousseeuw, 1987) – a widely used measure of how similar a member is to its own cluster (homogeneity within a cluster) compared to other clusters (heterogeneity across clusters ) when the ground-truth cluster labels are

---

[10]https://github.com/ratschlab/SOM-VAE

not available – to assess the identified clusters. Formally, the SI for a subsequence $\mathbf{x}_{1:t}^n \in \mathcal{C}^k$ can be given as follows:

$$SI(n) = \frac{b(n) - a(n)}{\max\left(a(n), b(n)\right)} \tag{12}$$

where $a(n) = \frac{1}{|\mathcal{C}^k|-1} \sum_{m \neq n} \|y_t^n - y_t^m\|_1$ and $b(n) = \min_{k' \neq k} \frac{1}{|\mathcal{C}^{k'}|} \sum_{m \in \mathcal{C}^{k'}} \|y_t^n - y_t^m\|_1$. Here, we used the L1-distance between the ground-truth labels of the future outcomes of interest since our goal is to group input subsequences with similar future outcomes. To control the number of identified clusters (i.e., the activated clusters) of our method, we set $\beta = 0$ (since the embedding separation loss in (5) controls the activation of clusters) and reported the performance by increasing the number of possible clusters $K$.

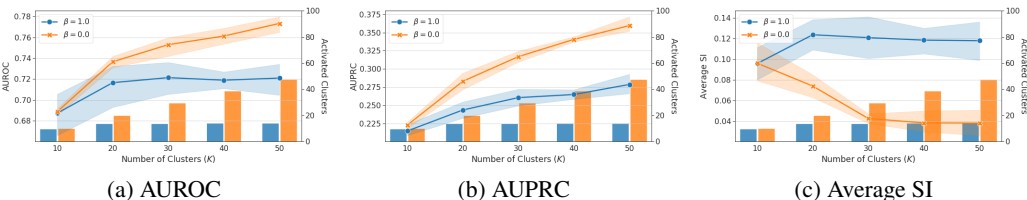

| (a) AUROC | (b) AUPRC | (c) Average SI |
|:---:|:---:|:---:|

Figure 5: AUROC, AUPRC, and average SI (mean and 95% confidence interval) and the number of activated clusters with various $K$.

As can be seen in Figure 5, the prediction performance increased with an increased number of identified clusters due to the higher diversity to represent the label distribution while making the identified clusters less interpretable (i.e., the cohesion and separation among clusters become ambiguous as seen in the low average SI). On the other hand, when we set $\beta = 1.0$ (which is selected based on the validation loss in 3), our method consistently identified a similar number of clusters for $K > 20$, i.e., 13.8 on average, in a data-driven fashion and provided slightly reduced prediction performance with significantly better interpretability, i.e., the average SI 0.120 on average. This highlights the usefulness of (5) which helps to identify clusters with different label distributions.

Table 7: The comorbidities developed in the next year for the 12 identified clusters. The values in parentheses indicate the corresponding frequency.

| Clusters | Comorbidities and Frequencies | | | |
|---|---|---|---|---|
| Cluster 0 | Diabetes (0.85) | Liver Enzymes (0.21) | Arthropathy (0.14) | Depression (0.10) |
| | Hypertens (0.08) | Osteopenia (0.07) | Intest. Obstruction (0.07) | Cirrhosis (0.04) |
| | ABPA (0.04) | Liver Disease (0.04) | Osteoporosis (0.03) | Hearing Loss (0.03) |
| | Asthma (0.02) | Kidney Stones (0.01) | Bone fracture (0.01) | Hemoptysis (0.01) |
| | Pancreatitis (0.01) | Cancer (0.00) | Gall bladder (0.00) | Colonic stricture (0.00) |
| | GI bleed – no var. (0.00) | GI bleed – var. (0.00) | | |
| Cluster 1 | Liver Enzymes (0.09) | Arthropathy (0.08) | Depression (0.07) | Intest. Obstruction (0.06) |
| | Diabetes (0.06) | Osteopenia (0.05) | ABPA (0.04) | Asthma (0.03) |
| | Liver Disease (0.03) | Hearing Loss (0.03) | Osteoporosis (0.02) | Pancreatitis (0.02) |
| | Kidney Stones (0.02) | Hypertension (0.01) | Cirrhosis (0.01) | Gall bladder (0.01) |
| | Cancer (0.01) | Hemoptysis (0.00) | Bone fracture (0.00) | Colonic stricture (0.00) |
| | GI bleed – no var. (0.00) | GI bleed – var. (0.00) | | |
| Cluster 2 | ABPA (0.77) | Osteopenia (0.21) | Intest. Obstruction (0.11) | Hearing Loss (0.10) |
| | Liver Enzymes (0.07) | Diabetes (0.06) | Depression (0.05) | Pancreatitis (0.05) |
| | Liver Disease (0.04) | Arthropathy (0.04) | Asthma (0.03) | Bone fracture (0.02) |
| | Osteoporosis (0.02) | Hypertension (0.01) | Cancer (0.01) | Cirrhosis (0.01) |
| | Kidney Stones (0.01) | Gall bladder (0.01) | Hemoptysis (0.00) | Colonic stricture (0.00) |
| | GI bleed – no var. (0.00) | GI bleed – var. (0.00) | | |
| Cluster 3 | Asthma (0.89) | Liver Disease (0.87) | Diabetes (0.29) | Osteopenia (0.28) |
| | Liver Enzymes (0.24) | ABPA (0.15) | Osteoporosis (0.11) | Hearing Loss (0.06) |
| | Arthropathy (0.05) | Intest. Obstruction (0.05) | Depression (0.04) | Hypertension (0.03) |
| | Cirrhosis (0.02) | Kidney Stones (0.02) | Pancreatitis (0.02) | Gall bladder (0.02) |
| | Cancer (0.01) | Bone fracture (0.00) | Hemoptysis (0.00) | Colonic stricture (0.00) |
| | GI bleed – no var. (0.00) | GI bleed – var. (0.00) | | |
| Cluster 4 | Osteoporosis (0.76) | Diabetes (0.43) | Arthropathy (0.20) | Liver Enzymes (0.18) |
| | Osteopenia (0.15) | Depression (0.13) | Intest. Obstruction (0.11) | ABPA (0.11) |
| | Hearing Loss (0.09) | Liver Disease (0.08) | Hypertension (0.07) | Cirrhosis (0.07) |
| | Kidney Stones (0.03) | Asthma (0.02) | Hemoptysis (0.02) | Bone fracture (0.02) |
| | Gall bladder (0.01) | Pancreatitis (0.01) | Cancer (0.00) | Colonic stricture (0.00) |
| | GI bleed – no var. (0.00) | GI bleed – var. (0.00) | | |
| Cluster 5 | Asthma (0.88) | Diabetes (0.81) | Osteopenia (0.28) | ABPA (0.24) |
| | Liver Enzymes (0.22) | Depression (0.15) | Osteoporosis (0.14) | Intest. Obstruction (0.12) |
| | Hypertension (0.10) | Cirrhosis (0.10) | Liver Disease (0.09) | Arthropathy (0.08) |
| | Bone fracture (0.01) | Hemoptysis (0.01) | Pancreatitis (0.01) | Hearing Loss (0.01) |
| | Cancer (0.01) | Kidney Stones (0.01) | GI bleed – var. (0.01) | Gall bladder (0.00) |
| | Colonic stricture (0.00) | GI bleed – no var. (0.00) | | |
| Cluster 6 | Liver Disease (0.85) | Liver Enzymes (0.37) | Osteopenia (0.27) | ABPA (0.09) |
| | Arthropathy (0.07) | Diabetes (0.06) | Intest. Obstruction (0.06) | Osteoporosis (0.05) |
| | Depression (0.03) | Asthma (0.03) | Hearing Loss (0.03) | Cirrhosis (0.02) |
| | Hemoptysis (0.02) | Hypertension (0.01) | Kidney Stones (0.01) | Pancreatitis (0.00) |
| | Gall bladder (0.00) | Bone fracture (0.00) | Cancer (0.00) | Colonic stricture (0.00) |
| | GI bleed – no var. (0.00) | GI bleed – var. (0.00) | | |
| Cluster 7 | ABPA (0.83) | Diabetes (0.78) | Osteopenia (0.25) | Osteoporosis (0.24) |
| | Liver Enzymes (0.15) | Intest. Obstruction (0.12) | Liver Disease (0.09) | Hypertension (0.07) |
| | Hearing Loss (0.07) | Arthropathy (0.06) | Depression (0.04) | Cirrhosis (0.02) |
| | Asthma (0.01) | Bone fracture (0.01) | Kidney Stones (0.01) | Hemoptysis (0.01) |
| | Cancer (0.00) | Pancreatitis (0.00) | Gall bladder (0.00) | Colonic stricture (0.00) |
| | GI bleed – no var. (0.00) | GI bleed – var. (0.00) | | |
| Cluster 8 | Diabetes (0.94) | Liver Disease (0.83) | Liver Enzymes (0.43) | Osteopenia (0.30) |
| | Hearing Loss (0.11) | Osteoporosis (0.10) | Intest. Obstruction (0.09) | Cirrhosis (0.08) |
| | Depression (0.08) | ABPA (0.07) | Arthropathy (0.06) | Hypertension (0.05) |
| | Kidney Stones (0.05) | Asthma (0.02) | Hemoptysis (0.01) | Bone fracture (0.01) |
| | Cancer (0.00) | Pancreatitis (0.00) | Gall bladder (0.00) | Colonic stricture (0.00) |
| | GI bleed – no var. (0.00) | GI bleed – var. (0.00) | | |
| Cluster 9 | Asthma (0.89) | Osteopenia (0.26) | ABPA (0.19) | Arthropathy (0.14) |
| | Intest. Obstruction (0.11) | Depression (0.08) | Osteoporosis (0.08) | Diabetes (0.06) |
| | Liver Enzymes (0.06) | Hemoptysis (0.03) | Hypertension (0.02) | Liver Disease (0.02) |
| | Pancreatitis (0.02) | Bone fracture (0.01) | Cancer (0.01) | Cirrhosis (0.01) |
| | Gall bladder (0.01) | Hearing Loss (0.01) | Kidney Stones (0.00) | Colonic stricture (0.00) |
| | GI bleed – no var. (0.00) | GI bleed – var. (0.00) | | |
| Cluster 10 | Osteopenia (0.82) | Diabetes (0.81) | Arthropathy (0.23) | Depression (0.19) |
| | Liver Enzymes (0.18) | Hypertension (0.16) | Hearing Loss (0.10) | Liver Disease (0.10) |
| | Osteoporosis (0.10) | Intest. Obstruction (0.09) | ABPA (0.09) | Kidney Stones (0.07) |
| | Cirrhosis (0.05) | Asthma (0.01) | Cancer (0.00) | GI bleed – var. (0.00) |
| | Bone fracture (0.00) | Hemoptysis (0.00) | Pancreatitis (0.00) | Gall bladder (0.00) |
| | Colonic stricture (0.00) | GI bleed – no var. (0.00) | | |
| Cluster 11 | Osteopenia (0.77) | Liver Enzymes (0.18) | Arthropathy (0.12) | Depression (0.09) |
| | Hypertension (0.06) | Diabetes (0.06) | Hearing Loss (0.06) | ABPA (0.05) |
| | Liver Disease (0.05) | Osteoporosis (0.04) | Intest. Obstruction (0.04) | Cirrhosis (0.02) |
| | Asthma (0.02) | Pancreatitis (0.02) | Bone fracture (0.01) | Cancer (0.01) |
| | Kidney Stones (0.00) | Gall bladder (0.00) | Colonic stricture (0.00) | Hemoptysis (0.00) |
| | GI bleed – no var. (0.00) | GI bleed – var. (0.00) | | |

# F    PSEUDO-CODE OF AC-TPC

As illustrated in Section 3.2, AC-TPC is trained in an iterative fashion. We provide the pseudo-code for optimizing our model in Algorithm 1 and that for initializing the parameters in Algorithm 2.

---

**Algorithm 1** Pseudo-code for Optimizing AC-TPC

---

**Input:** Dataset $\mathcal{D} = \{(\mathbf{x}_t^n, y_t^n)_{t=1}^{T^n}\}_{n=1}^{N}$, number of clusters $K$, coefficients $(\alpha, \beta)$,
      learning rate $(\eta_A, \eta_C, \eta_E)$, mini-batch size $n_{mb}$, and update step $M$
**Output:** AC-TPC parameters $(\theta, \psi, \phi)$ and the embedding dictionary $\mathcal{E}$
Initialize parameters $(\theta, \psi, \phi)$ and the embedding dictionary $\mathcal{E}$ via `Algorithm 2`

**repeat**
    ***Optimize the Encoder, Selector, and Predictor***
    **for** $m = 1, \cdots, M$ **do**
        Sample a mini-batch of $n_{mb}$ data samples: $\{(\mathbf{x}_t^n, y_t^n)_{t=1}^{T^n}\}_{n=1}^{n_{mb}} \sim \mathcal{D}$
        **for** $n = 1, \cdots, n_{mb}$ **do**
            Calculate the assignment probability:   $\pi_t^n = [\pi_t^n(1) \cdots \pi_t^n(K)] \leftarrow h_\psi(f_\theta(\mathbf{x}_{1:t}^n))$
            Draw the cluster assignment:   $s_t^n \sim Cat(\pi_t^n)$
            Calculate the label distributions:   $\bar{y}_t^n \leftarrow g_\phi(\mathbf{e}(s_t^n))$ and $\hat{y}_t^n \leftarrow g_\phi(f_\theta(\mathbf{x}_{1:t}^n))$
        **end for**
        Update the encoder $f_\theta$ and selector $h_\psi$:

$$\theta \leftarrow \theta - \eta_A \left( \frac{1}{n_{mb}} \sum_{n=1}^{n_{mb}} \sum_{t=1}^{T^n} \ell_1(y_t^n, \bar{y}_t^n) \nabla_\theta \log \pi_t^n(s_t^n) - \alpha \nabla_\theta \sum_{k=1}^{K} \pi_t^n(k) \log \pi_t^n(k) \right)$$

$$\psi \leftarrow \psi - \eta_A \left( \frac{1}{n_{mb}} \sum_{n=1}^{n_{mb}} \sum_{t=1}^{T^n} \ell_1(y_t^n, \bar{y}_t^n) \nabla_\psi \log \pi_t^n(s_t^n) - \alpha \nabla_\psi \sum_{k=1}^{K} \pi_t^n(k) \log \pi_t^n(k) \right)$$

        Update the predictor $g_\phi$:

$$\phi \leftarrow \phi - \eta_C \frac{1}{n_{mb}} \sum_{n=1}^{n_{mb}} \sum_{t=1}^{T^n} \nabla_\phi \ell_1(y_t^n, \bar{y}_t^n)$$

    **end for**

    ***Optimize the Cluster Centroids***
    **for** $m = 1, \cdots, M$ **do**
        Sample a mini-batch of $n_{mb}$ data samples: $\{(\mathbf{x}_t^n, y_t^n)_{t=1}^{T^n}\}_{n=1}^{n_{mb}} \sim \mathcal{D}$
        **for** $n = 1, \cdots, n_{mb}$ **do**
            Calculate the assignment probability:   $\pi_t^n = [\pi_t^n(1) \cdots \pi_t^n(K)] \leftarrow h_\psi(f_\theta(\mathbf{x}_{1:t}^n))$
            Draw the cluster assignment:   $s_t^n \sim Cat(\pi_t^n)$
            Calculate the label distributions:   $\bar{y}_t^n \leftarrow g_\phi(\mathbf{e}(s_t^n))$
        **end for**
        **for** $k = 1, \cdots, K$ **do**
            Update the embeddings $\mathbf{e}(k)$:

$$\mathbf{e}(k) \leftarrow \mathbf{e}(k) - \eta_E \left( \frac{1}{n_{mb}} \sum_{n=1}^{n_{mb}} \sum_{t=1}^{T^n} \nabla_{\mathbf{e}(k)} \ell_1(y_t^n, \bar{y}_t^n) - \gamma \sum_{\substack{k'=1 \\ k' \neq k}}^{K} \nabla_{\mathbf{e}(k)} \ell_1\big(g_\phi(\mathbf{e}(k)), g_\phi(\mathbf{e}(k'))\big) \right)$$

        **end for**
        Update the embedding dictionary:   $\mathcal{E} \leftarrow \{\mathbf{e}(1), \ldots \mathbf{e}(K)\}$
    **end for**
**until** convergence

---

---

**Algorithm 2** Pseudo-code for pre-training AC-TPC

---

**Input:** Dataset $\mathcal{D} = \{(\mathbf{x}_t^n, y_t^n)_{t=1}^{T^n}\}_{n=1}^N$, number of clusters $K$, learning rate $\eta$, mini-batch size $n_{mb}$
**Output:** AC-TPC parameters $(\theta, \psi, \phi)$ and the embedding dictionary $\mathcal{E}$
Initialize parameters $(\theta, \psi, \phi)$ via Xavier Initializer

*__Pre-train the Encoder and Predictor__*
**repeat**
    Sample a mini-batch of $n_{mb}$ data samples: $\{(\mathbf{x}_t^n, y_t^n)_{t=1}^{T^n}\}_{n=1}^{n_{mb}} \sim \mathcal{D}$
    **for** $n = 1, \cdots, n_{mb}$ **do**
        Calculate the label distributions:   $\hat{y}_t^n \leftarrow g_\phi(f_\theta(\mathbf{x}_{1:t}^n))$
    **end for**

$$\theta \leftarrow \theta - \eta \frac{1}{n_{mb}} \sum_{n=1}^{n_{mb}} \sum_{t=1}^{T^n} \nabla_\theta \ell_1(y_t^n, \hat{y}_t^n) \qquad \phi \leftarrow \phi - \eta \frac{1}{n_{mb}} \sum_{n=1}^{n_{mb}} \sum_{t=1}^{T^n} \nabla_\phi \ell_1(y_t^n, \hat{y}_t^n)$$

**until** convergence

*__Initialize the Cluster Centroids__*
Calculate the embedding dictionary $\mathcal{E}$ and initial cluster assignments $c_t^n$

$$\mathcal{E}, \{\{c_t^n\}_{t=1}^{T^n}\}_{n=1}^N \leftarrow \texttt{K-means}(\{\{\mathbf{z}_t^n\}_{t=1}^{T^n}\}_{n=1}^N, K)$$

*__Pre-train the Selector__*
**repeat**
    Sample a mini-batch of $n_{mb}$ data samples: $\{(\mathbf{x}_t^n, y_t^n)_{t=1}^{T^n}\}_{n=1}^{n_{mb}} \sim \mathcal{D}$
    **for** $n = 1, \cdots, n_{mb}$ **do**
        Calculate the assignment probability:   $\pi_t^n = [\pi_t^n(1) \cdots \pi_t^n(K)] \leftarrow h_\psi(f_\theta(\mathbf{x}_{1:t}^n))$
    **end for**
    Update the selector $h_\psi$:

$$\psi \leftarrow \psi + \eta \frac{1}{n_{mb}} \sum_{n=1}^{n_{mb}} \sum_{t=1}^{T^n} \sum_{k=1}^{K} c_t^n(k) \log \pi_t^n(k)$$

**until** convergence

---

