# OpenReview forum: "Actor-Critic Approach for Temporal Predictive Clustering"
_ICLR.cc/2020/Conference — Reject_

### Official Review · AnonReviewer2 · 2019-10-24
**Official Blind Review #2**

**Rating:** 3

**Review:**

Summary:
This work proposes a method to perform clustering on a time-series data for prediction purposes, unlike the classical clustering where it is done in an unsupervised manner. The authors use an encoder (RNN) to process the time-series medical records, a selector to sample the cluster label for each encoding, and a predictor to predict the labels based on the selected cluster centroids. Since the sampling process prevents the authors from using back-prop, they employ an actor-critic method.

Strengths:
- Although some would argue otherwise, patient similarity has some promise to be useful in clinical practice.
- The proposed method clearly outperformed various baselines in terms of clustering (Table 1).
- Table 3 and Figure 3 show that the proposed method can capture heterogeneous subgroups of the dataset.

Concerns:
- I'm not a clustering expert, but I'm skeptical this is the first work to combine clustering and supervised prediction using an RL technique.
- It is unclear what it means to train the embedding dictionary. Are there trainable parameters in the embedding dictionary? It seems that all it does is calculate the mean of the z_t's (i.e. centroid) in each cluster. Or do you take the centroid embeddings and put that through a feed-forward network of some sort?
- The effect of Embedding Separation Loss (Eq.5) seems quite limited. According to Table 2, it doesn't seem to help much. And contrary to the authors' claim, using \beta increase the number of activated clusters from 8 to 8.4.
- Most importantly, the central theme of this work is combining clustering with prediction labels for the downstream prediction task. But the authors do not compare the prediction performance of the proposed method with other clustering method or "patient similarity" methods, or even simple supervised models. The only prediction performance metric to be found is Table 2 from the ablation study.

**Experience Assessment:**

I have published one or two papers in this area.

**Review Assessment: Checking Correctness Of Derivations And Theory:**

I carefully checked the derivations and theory.

**Review Assessment: Checking Correctness Of Experiments:**

I carefully checked the experiments.

**Review Assessment: Thoroughness In Paper Reading:**

I read the paper thoroughly.

---

> ### Author Response · Authors · 2019-11-14
> **Response to Reviewer#2**
>
> We thank the reviewer for the valuable comments.
>
> A1. To our best knowledge, this paper is the first to apply reinforcement learning for identifying predictive clusters. Indeed, there exist some studies that explored clustering with reinforcement learning, such as to select a better initialization for K-means [A] and to design efficient feature space [B]. However, previous works mainly focused on conventional clustering problems without accounting for observed outcome labels of interest.
>
> A2. The embedding dictionary consists of $K$ embedding vectors in the latent space, i.e., $e(1), … , e(K)$ where these embedding vectors are updated by minimizing the loss function in (9) based on the stochastic gradient descent method. This is doable because both terms in (9) are defined as a function of the predictor outputs taking the embedding vectors as input; this makes (9) differentiable with respect to the embedding vectors. Please refer to Algorithm 1 in Appendix F to see the formal descriptions of how we update the embedding vectors.
>
> A3. We inadvertently mistyped the values on the number of activated clusters for $\alpha=3.0, \beta=0$ and $\alpha=3.0, \beta=1.0$; the correct number of activated clusters is 8.4 with $\beta=0$ and 8 with $\beta=1.0$. We will correct the typo in Table 2 in the revised manuscript. Besides, please refer to A2 to Reviewer #1 to see how the embedding separation loss in (5) plays a significant role in identifying the number of clusters in a data-driven fashion.
>
> A4. As the reviewer suggested, we assessed the prognostic performance of our method and the clustering benchmarks which incorporate the label information during training – that are KM-E2P ($\mathcal{Z}$), KM-E2P ($\mathcal{Y}$), DCN-E2P, and SOM-VAE-P. For the prognostic performance, we evaluated AUROC and AURPC based on the ground-truth binary outcomes ($y_{t}$) and the cluster-specific outcome predictions that are calculated specifically to each clustering method:
>     - For KM-E2P ($\mathcal{Z}$) and DCN-E2P, the cluster-specific outcome predictions are the predictor outcomes taking the average latent representations per cluster (i.e., the K-means centroids in the latent space) as input.
>     - For KM-E2P ($\mathcal{Y}$), the cluster-specific outcome predictions are the average predictor outcomes per cluster (i.e., the K-means centroids in the outcome space).
>     - For SOM-VAE-P and our method, the cluster-specific outcome predictions are the predictor outcomes (i.e., $\bar{y}$) taking the assigned embedding vectors as input.
> Below, we reported the AUROC and AUPRC for UKCF and ADNI; our method outperformed all the tested benchmarks in both performance metrics. We will update Table 1 in the revised manuscript accordingly.
>
> ---------------------------------------------------------------------------------------------------
>                        |UKCF                                         | ADNI
> Methods       |--------------------------------------------------------------------------------
>                        | AUROC          | AUPRC           | AUROC           | AUPRC
> ---------------------------------------------------------------------------------------------------
> KM-E2P(Z)    | 0.726 $\pm$ 0.01 | 0.425 $\pm$ 0.02 | 0.707 $\pm$ 0.01 | 0.509 $\pm$ 0.01
> KM-E2P(Y)    | 0.807 $\pm$ 0.00 | 0.514 $\pm$ 0.01 | 0.756 $\pm$ 0.04 | 0.503 $\pm$ 0.04
> DCN-E2P      | 0.772 $\pm$ 0.03 | 0.487 $\pm$ 0.03 | 0.721 $\pm$ 0.03 | 0.509 $\pm$ 0.03
> SOM-VAE-P  | 0.754 $\pm$ 0.05 | 0.331 $\pm$ 0.07 | 0.597 $\pm$ 0.10 | 0.376 $\pm$ 0.05
> Proposed     | 0.843 $\pm$ 0.01 | 0.605 $\pm$ 0.01 | 0.768 $\pm$ 0.02 | 0.515 $\pm$ 0.02
> ---------------------------------------------------------------------------------------------------
>
> References:
> [A] S. Bose and M. Huber, “Semi-Unsupervised Clustering Using Reinforcement Learning,” AAAI 2016.
> [B] W. Barbakh and C. Fyfe, “Clustering with Reinforcement Learning,” IDEAL 2007

---

### Official Review · AnonReviewer1 · 2019-10-24
**Official Blind Review #1**

**Rating:** 3

**Review:**

This paper proposes a temporal clustering algorithm for the medical domain. The main advantage of the proposed method is that it uses supervised information for temporal clustering. The proposed method is evaluated on two real-world datasets and showed improvements against a few other temporal clustering methods.

Detailed Comments:

Methodology:
The actor-critic part of the loss is really just a type of policy gradient. There is no estimation of a value function in Eqn. 7 or anywhere to be found in Appendix B. This is very misleading on the part of the authors because policy grad and AC are very different algorithms. AC type of algorithms provide some variance reduction mechanisms for the classical policy gradient (not done here), but they require additional work to estimate a value function for future trajectories that incur bias. If the authors claim a real AC algorithm for the predictive clustering loss, then some justification for bias-variance tradeoffs should be mentioned instead of attributing it to tuning the hyperparameters between the losses.

Perhaps a bigger issue is that there is a general tradeoff between reconstructing the time-series (the unsupervised learning portion) vs. predictive performance that is common to predictive clustering problems -- it is not addressed here. One can increase the performance in one (in this case prediction accuracy) while sacrificing the other.

In the extreme case, one can just tune the hyperparameter of the loss function to turn this into a pure supervised learning problem while sacrificing the capacity of the embedding representations to have any meaning (e.g., to reconstruct time-series [SOM-VAE] or predict future ones). The authors did not really propose a systematic way to control this tradeoff, nor did they provide some experiments to show how well the embeddings can be used to recover the temporal patterns.

Experiments:
The baseline experiments are rather weak. For example, 3 out of the 4 models (DTW, DCN extensions and SOM-VAE) were all unsupervised learning techniques that are not designed to take into account label information. This is especially true for comparison against SOM-VAE (which the authors called state of the art for the proposed problem). SOM-VAE is actually less intended for prognostication than the likes of Baytas et al. (2017) and Madiraju et al. (2018) and provides less informative baselines. The authors should also provide more details regarding how these ``extensions'' are done for things like DCN and K-means clustering on deep neural networks.

**Experience Assessment:**

I have read many papers in this area.

**Review Assessment: Checking Correctness Of Derivations And Theory:**

N/A

**Review Assessment: Checking Correctness Of Experiments:**

I carefully checked the experiments.

**Review Assessment: Thoroughness In Paper Reading:**

I read the paper thoroughly.

---

> ### Author Response · Authors · 2019-11-14
> **Response to Reviewer#1 - Part 1/2**
>
> We thank the reviewer for valuable comments.
>
> A1. In our problem formulation, we defined the value function (also known as the expected total reward) as an immediate reward of selecting a cluster. More specifically, at each time stamp $t$, the input subsequence $x_{1:t}$ is the state, the cluster assignment $s_{t}$ given $x_{1:t}$ is the action given the policy $\pi_{t}$, and $\ell_{1}(y_{t}, \bar{y}_{t})$ is the value function which assesses how well the label distribution of a member can be represented by the label distribution of the assigned cluster. Since the future trajectory of state-action pairs is not changed by the current cluster assignment, we only consider the immediate reward of selecting a cluster to define the value function. Similar training procedures using reinforcement learning have been proposed to overcome sampling processes for feature selection [A] (actor-critic) and question and answering [B] (REINFORCE); in both cases, the action does not change the future state-action pairs and, thus, the value function is defined by the immediate reward.
> Given the description above, the cluster assignment is chosen by the “actor” (i.e., the encoder $f_{\theta}$ + selector $h_{\psi}$) and the value function $\ell_{1}(y_{t}, \bar{y}_{t})$ is estimated based on the output of the “critic” (i.e. the predictor $g_{\phi}$); please note that $\bar{y}_{t}$ is the output of the predictor given the embedding of the assigned cluster, i.e. $e(s_{t})$. Since we iteratively update the actor for better cluster assignments and the critic for better estimation of the immediate reward, we believe that our training procedure belongs to actor-critic methods rather than policy gradient methods (that do not necessarily update the critic).
>
> As the reviewer points out, actor-critic methods, in general, use a value function (which is a discounted sum of future rewards) that is defined based on the future trajectory of state-action pairs which in turn makes the training susceptible to “noisy” estimations of the future rewards. Contrarily, the value function used in our method is an immediate reward -- that is defined based only on the current action (as described above) -- and thus the training is less likely to suffer from “noisy” estimations. In addition, pretraining the network as described in Algorithm 2 in Appendix F encourages the actor and the critic to accurately predict the label distribution which leads to a less noisy estimation of the value function at the beginning of the training.
>
> We will clarify the description in the revised manuscript accordingly.
>
>
> A3. As the reviewer points out, the original versions of DTW, DCN, and SOM-VAE do not incorporate label information. This is EXACTLY the reason why we additionally compared our method against variations of DCN (denoted as DCN-E2P) and SOM-VAE (denoted as SOM-VAE-P) that take label information during training. Thus, the amount of information provided for both training our method and training these benchmarks are the same. More specifically, we replaced the decoder of DCN and SOM-VAE with the predictor and trained the networks based on the prediction loss in (10) in order to provide label prediction instead of reconstructing the input.
> By doing so, DCN-E2P and SOM-VAE-P find latent representations -- in which clustering (or SOM) is conducted -- that preserve information for label prediction. In the revised manuscript, we will clarify the descriptions of these variations and add illustrations of the network architectures (for all benchmarks) in Appendix D.
>
>
> References:
> [A] J. Yoon et al., “INVASE: Instance-Wise Variable Selection Using Neural Networks,” ICLR 2019.
> [B] N.R. Ke et al., “Focused Hierarchical RNNs for Conditional Sequence Processing,” ICML 2018.
> [C] P. J. Rousseeuw, “Silhouettes: a graphical aid to the interpretation and validation of cluster analysis,” Computational and Applied Mathematics, 1987

---

> ### Author Response · Authors · 2019-11-14
> **Response to Reviewer#1 - Part 2/2**
>
> A2. We thank the reviewer for pointing such an important issue in the predictive clustering and for providing an opportunity to highlight our contributions. In predictive clustering, it is true that the trade-off between the clustering performance (for better interpretability), which quantifies how the data samples are homogeneous within each cluster and heterogeneous across clusters with respect to the future outcomes of interest, and the prediction performance is a common issue. It is worth highlighting that predictive clustering is an unsupervised task of finding groups of samples with similar future outcomes of interest(i.e., the label for true clusters is not available); therefore, it is not necessary to incorporate reconstruction of the original input.
>
> The most important parameter that governs this trade-off is the number of clusters. More specifically, increasing the number of clusters will make the predictive clusters have higher diversity to represent the output distribution and, thus, will increase the prediction performance while decreasing the clustering performance. One extreme example is that there are as many clusters as data samples which will make the identified clusters fully individualized; as a consequence, each cluster will lose the interpretability as it no longer groups similar data samples.
>
> To highlight this trade-off, we conducted extensive experiments under the same experimental setup with that of Section 5.6 where our aim is to identify underlying (unknown) clusters when the future outcome of interest is high-dimensional. For the performance measures, we utilized the AUROC and AUPRC to assess the prediction performance, and utilized the Silhouette index [C] -- a widely used measure of how similar a member is to its own cluster (homogeneity within a cluster) compared to other clusters (heterogeneity across clusters ) when the ground-truth cluster labels are not available -- to assess the identified clusters. Here, we used the L1-distance between the ground-truth output labels to compute the Silhouette index since our goal is to group input subsequences with similar future outcomes. The formal description will be added in the appendix of the revised manuscript. To control the number of identified clusters (i.e., the activated clusters) of our method, we set $\beta=0$ (since the embedding separation loss in (5) controls the activation of clusters) and reported the performance by increasing the number of possible clusters $K$ (which is the dimension of the softmax output layer of the selector).
>
> As can be seen in the table below, the prediction performance increased with an increased number of clusters due to the higher diversity to represent the label distribution while having the identified clusters less interpretable (i.e., the cohesion and separation among clusters become ambiguous). On the other hand, when we set $\beta=1.0$ (which is selected based on the validation loss in (3)), our method consistently identified a similar number of clusters for $K > 20$, i.e., 13.8 on average, in a data-driven fashion and provided slightly reduced prediction performance with significantly better interpretability, i.e., the Silhouette index 0.121 on average. This highlights the usefulness of (5) which helps to identify clusters to have different label distributions. Detailed results will be added to the revised manuscript.
>
> -------------------------------------------------------------------------------------------------------------------------------
>                  |Metrics                    |K=10            |K=20            |K=30            |K=40            |K=50
> -------------------------------------------------------------------------------------------------------------------------------
> $\beta =0$      |AUROC                    |0.689$\pm$0.01|0.737$\pm$0.01|0.753$\pm$0.01|0.761$\pm$0.01|0.773$\pm$0.01
>                  |AUPRC                     |0.223$\pm$0.01|0.283$\pm$0.01|0.317$\pm$0.01|0.341$\pm$0.01|0.359$\pm$0.01
>                  |Silhouette Index    |0.096$\pm$0.02|0.074$\pm$0.01|0.043$\pm$0.01|0.038$\pm$0.01|0.038$\pm$0.02
>                  |Activated Clusters |10                |19.8             |29.4              |38.6             |47.6
> -------------------------------------------------------------------------------------------------------------------------------
> $\beta=1$      |AUROC                    |0.687$\pm$0.01|0.716$\pm$0.01|0.721$\pm$0.01|0.719$\pm$0.01|0.722$\pm$0.02
>                  |AUPRC                     |0.215$\pm$0.01|0.244$\pm$0.01|0.261$\pm$0.01|0.251$\pm$0.01|0.269$\pm$0.01
>                  |Silhouette Index    |0.096$\pm$0.02|0.124$\pm$0.01|0.121$\pm$0.01|0.119$\pm$0.02|0.118$\pm$0.02
>                  |Activated Clusters | 9.6              |13.6              |13.7             |13.8             |14.1
> -------------------------------------------------------------------------------------------------------------------------------

---

### Official Review · AnonReviewer3 · 2019-10-26
**Official Blind Review #3**

**Rating:** 6

**Review:**

The authors propose a clustering approach for time series that encourages instances with similar time profiles to be clustered together. The approach consists of three modules: an encoder, a cluster assigner and a (future outcome) predictor, all specified as neural networks.

The objective of the model is to produce cluster embeddings that are as informative of the outcomes as possible, while not being a direct function of covariates. Note that (2) may look misleading because it indicates that the outcome is a function of the cluster assignment, however, it does not show that the assignment is indeed a function of the covariates.

It is not entirely clear how a patient is assigned to a cluster provided that cluster assignments are a function of time.

It is desirable that performance metrics do not seem affected by the unknown number of clusters, however, this makes for difficult to interpret clusters. More so in practice when the number of identified clusters is a function of the model architecture and hyperparameters (\alpha and \beta). Is the number of clusters selected by cross-validation and if so, what performance metric is used to select the best choice?

In Table 3 for UKCF with 3 comorbidities, how are AUROC and AUPRC evaluated provided these are binary predictions?

**Experience Assessment:**

I have published one or two papers in this area.

**Review Assessment: Checking Correctness Of Derivations And Theory:**

I carefully checked the derivations and theory.

**Review Assessment: Checking Correctness Of Experiments:**

I carefully checked the experiments.

**Review Assessment: Thoroughness In Paper Reading:**

I read the paper thoroughly.

---

> ### Author Response · Authors · 2019-11-14
> **Response to Reviewer#3**
>
> We thank the reviewer for the valuable comments.
>
> A1. In the probabilistic definition of the KL-divergence in (2), it is not necessary to explicitly denote the dependency between $S_{t}$ (i.e., the random variable for cluster assignment at time $t$) and $X_{1:t}$ (i.e., the random variable for the input subsequence at time $t$). Instead, the dependency between the random variables is further specified in (3) in terms of their realizations. More specifically, $\bar{y}_{t}$ in (3) is a function of $s_{t}$ that is drawn from a categorical distribution whose probability of each category is defined as a function of the input subsequence $x_{1:t}$. In the revised manuscript, we will clarify the description of the dependency between the two random variables in (2).
>
> A2. For a run-time (testing) example, suppose that we have a new patient whose time-series observations are given as $x_{1:T}$. Then, our method assigns cluster $s_{T}$ to this patient based on the input sequence $x_{1:T}$. When a new observation $x_{T+1}$ on this patient is collected at time $T+1$, we can update the cluster assignment of this patient to $s_{T+1}$ given the input sequence $x_{1:T+1}$.
>
> A3. We selected the hyperparameters of our network that give the minimum validation loss in (3). It is worth highlighting that, given the hyperparameters selected, our method identifies the number of clusters in a data-driven fashion; please refer to A2 to Reviewer #1 to see how the identified number of clusters remains consistent throughout different $K$. We will clarify this in the revised manuscript.
>
> A4. In the experiments, we know the ground-truth binary labels of the future outcomes of interest (that is, development of diabetes, ABPA, and intestinal obstruction). Thus, we can simply compute AUROC and AUPRC using the cluster-wise outcome predictions (i.e., $\bar{y}$) and the outcome labels (i.e., $y$). The AUROC and AURPC reported in Table 2 are averaged over the three comorbidities. Similarly, the ground-truth cluster label can be computed as a combination of the three binary outcome labels which makes $C=2^{3}=8$ as described in Section 5.4. We will clarify the description in the revised manuscript.

---

### Decision · Program_Chairs · 2019-12-19

**Decision:**

Reject

**Comment:**

This paper proposes a reinforcement learning approach to clustering time-series data. The reviewers had several questions related to clarity and concerns related to the novelty of the method, the connection to RL, and experimental results. While the authors were able to address some of these questions and concerns in the rebuttal, the reviewers believe that the paper is not quite ready for publication.